# Approximation Theory for Lipschitz Continuous Transformers

**Takashi Furuya** [* 1]  **Davide Murari** [* 2]  **Carola-Bibiane Schönlieb** [2]

## Abstract

Stability and robustness are critical for deploying Transformers in safety-sensitive settings. A principled way to enforce such behavior is to constrain the model's Lipschitz constant. However, approximation-theoretic guarantees for architectures that explicitly preserve Lipschitz continuity have yet to be established. In this work, we bridge this gap by introducing a class of gradient-descent-type in-context Transformers that are Lipschitz-continuous by construction. We realize both MLP and attention blocks as explicit Euler steps of negative gradient flows, ensuring inherent stability without sacrificing expressivity. We prove a universal approximation theorem for this class within a Lipschitz-constrained function space. Crucially, our analysis adopts a measure-theoretic formalism, interpreting Transformers as operators on probability measures, to yield approximation guarantees independent of token count. These results provide a rigorous theoretical foundation for the design of robust, Lipschitz continuous Transformer architectures.

## 1. Introduction

Transformers are at the forefront of contemporary machine learning, but are susceptible to adversarial examples (Gupta & Verma, 2023; Xu et al., 2023), and can be unstable to train (Liu et al., 2020; Davis et al., 2021; Qi et al., 2023). A mathematically sound remedy against these issues is to control the Lipschitz constant of the input-to-output map they realize.

Getting reliable Lipschitz control is structurally challenging in Transformers. Standard self-attention involves un-

---

*Equal contribution [1]Faculty of Life and Medical Sciences, Department of Biomedical Engineering, Doshisha University, RIKEN AIP [2]Department of Applied Mathematics and Theoretical Physics, University of Cambridge. Correspondence to: Takashi Furuya <tfuruya@mail.doshisha.ac.jp>.

*Proceedings of the $43^{rd}$ International Conference on Machine Learning*, Seoul, South Korea. PMLR 306, 2026. Copyright 2026 by the author(s).

bounded dot products of linear projections, rendering it non-globally Lipschitz continuous (Kim et al., 2021). These issues are further exacerbated by the residual connections, which do not yield tight bounds on their Lipschitz constants (Xu et al., 2023; Li et al., 2023). Since computing a map's Lipschitz constant is NP-hard (Virmaux & Scaman, 2018), controlling the Lipschitz constant of Transformers is typically achieved by modifying and constraining a new architecture. Existing methods typically rely on spectral regularization, modifications to the self-attention layer, and the weighting of residual layers.

Among the various Lipschitz constraints, the 1-Lipschitz case is particularly natural (Sherry et al., 2024; Murari et al., 2025; Prach et al., 2023; Béthune et al., 2022; Hasannasab et al., 2019; Xu et al., 2023; Anil et al., 2019; Delattre et al., 2026). First, it is sometimes a necessary modeling requirement: 1-Lipschitz function classes arise as the admissible critics to approximate the Wasserstein-1 distance (Arjovsky et al., 2017), and being 1-Lipschitz is the baseline condition needed to obtain contraction maps in fixed-point iterations and stability analyzes. Second, even when one ultimately cares about an $L$-Lipschitz model, restricting the focus to 1-Lipschitz maps is largely without loss of generality, since any $L$-Lipschitz map can be written as a simple rescaling of a 1-Lipschitz map (Béthune et al., 2022). This makes 1-Lipschitzness a convenient normalization since it fixes a canonical scale for sensitivity while retaining the essential approximation questions within a standardized function class.

Transformers can be viewed as *in-context maps* in the sense that they take as input both a context and a distinguished query (or query token) and output a prediction that depends on both. Requiring the resulting map to be 1-Lipschitz with respect to the query for each fixed context ensures nonexpansiveness with respect to the query, which is useful, for instance, in learned fixed-point schemes that require convergence guarantees. This parallels the context-free setting; see (Hasannasab et al., 2019; Sherry et al., 2024). In the in-context case, however, varying the context changes the induced query-to-output transformation, effectively yielding a family of maps parameterized by the context. We therefore also require Lipschitz continuity with respect to the context for each fixed query, to ensure stability and robustness under small perturbations of the context. In this work, we model

the context as a probability measure and control its effect via Lipschitz continuity with respect to the Wasserstein-1 distance.

We introduce a Transformer architecture that is provably 1-Lipschitz with respect to the query for any fixed context, and admits a finite Lipschitz constant as a function of the context. The 1-Lipschitz bound is independent of the context length, and our layer is defined in the more general setting where the context is a measure. Moreover, we theoretically analyze the approximation capabilities of these models, proving universal approximation for the class of 1-Lipschitz in-context maps on compact domains. To the best of our knowledge, this is the first theoretical investigation of the approximation properties of 1-Lipschitz in-context maps.

### 1.1. Related work

**Lipschitz-constrained neural networks** Lipschitz control has been pursued as a principled mechanism for stability, robustness, and regularization of neural networks. Such a control on the Lipschitz constant is usually achieved by constraining the operator norms of linear layers during training (Miyato et al., 2018; Gouk et al., 2021; Bungert et al., 2021; Trockman & Kolter, 2021), and encoding non-expansiveness structurally through architectural design (Cisse et al., 2017; Sherry et al., 2024; Meunier et al., 2021). Especially when new architectural choices are made, some study of the network expressivity is necessary. A growing body of papers is exploring the theoretical approximation properties of Lipschitz-constrained networks. (Anil et al., 2019; Neumayer et al., 2023) study 1-Lipschitz feedforward networks, and (Murari et al., 2025) 1-Lipschitz ResNets. Our work builds on results from Section 3 of (Murari et al., 2025), but significantly extends their theory to accommodate the context measure, which plays a crucial role in Transformers.

**Lipschitz-constrained Transformers.** Enforcing Lipschitz guarantees in Transformers is challenging because key components are not globally Lipschitz under standard formulations; in particular, dot-product self-attention is not globally Lipschitz on unbounded domains (Kim et al., 2021). Prior work pursues Lipschitz control through architectural changes, including replacing dot-product attention with Lipschitz-controlled variants (Kim et al., 2021; Xu et al., 2023), modifying normalization (e.g., CenterNorm in Lips-Former) (Qi et al., 2023), and reweighting residual branches to prevent depth-wise growth of sensitivity (Qi et al., 2023; Newhouse et al., 2025). A particular mention needs to be made for the recent paper (Delattre et al., 2026), where the authors construct a 1-Lipschitz self-attention layer starting from convex gradient flows, similarly to how we approach the layer design. Our paper distinguishes from (Delattre et al., 2026) in that we study the approximation properties

of our newly proposed models, and rely on explicit layers rather than proximal maps to define the self-attention layer. Complementary contributions sharpen component-wise analysis, such as tight Lipschitz constants for softmax (Nair, 2026), and develop training-time norm-enforcement methods that aim to retain accuracy while providing explicit bounds (Newhouse et al., 2025). A substantial body of work, especially in vision, leverages Lipschitz control for certified robustness of transformer models (Xu et al., 2023; Menon et al., 2025; Gupta & Verma, 2023).

**Approximation theory of Transformers** The expressivity of Transformer architectures has been studied extensively. The foundational work of (Yun et al., 2020) established that Transformers with sufficient depth and width are universal approximators of continuous permutation-equivariant sequence-to-sequence functions on compact domains. Subsequent research has refined these guarantees, investigating the specific roles of positional encodings and attention heads in recognizing formal languages (Bhattamishra et al., 2020). From a computational perspective, (Pérez et al., 2021) demonstrated that Transformers with hard attention are Turing complete under arbitrary precision assumptions.

More recently, the theoretical analysis has shifted toward the in-context learning setting. (Akyürek et al., 2023) and (von Oswald et al., 2023) show that Transformers can implement explicit learning algorithms, including gradient-descent-type updates, within their forward pass. Crucially, (Furuya et al., 2025) recently advanced this analysis by adopting a measure-theoretic formalism, proving that Transformers are universal in-context learners with approximation rates independent of the token count. Note that standard Transformer approximation results, such as those in (Furuya et al., 2025), do not control the Lipschitz constants, which may grow with the model complexity. Moreover, prior measure-theoretic formulations do not incorporate architectural Lipschitz constraints.

### 1.2. Contributions

We summarize our main contributions as follows.

- **Lipschitz continuous in-context Transformers.** We introduce a new class of in-context Transformer architectures in which both the MLP and attention blocks are interpreted as explicit Euler steps of negative gradient flows. This design yields gradient-descent-type updates and allows for enforcing Lipschitz continuity of the resulting input-to-output map. Such a map is 1-Lipschitz with respect to the input query.

- **Universal approximation under explicit Lipschitz control.** We prove a universal approximation theorem for the proposed Lipschitz continuous in-context

Transformers. To the best of our knowledge, this provides the first approximation-theoretic guarantee for Transformer architectures that explicitly preserve Lipschitz continuity. Existing works primarily focus on analyzing or estimating Lipschitz properties rather than establishing approximation results.

- **Measure-theoretic in-context formulation with token-independent guarantees.** We cast in-context Transformers as operators acting on probability measures, and derive approximation guarantees that are independent of the number of tokens. The primary technical tool is a modification of the Restricted Stone–Weierstrass Theorem tailored to the measure-theoretic in-context setting. Since the Restricted Stone–Weierstrass framework is inherently constructive, our theoretical analysis also provides insights that are relevant for practical architectural design.

**Conflict of Interest Disclosure**   The authors declare no financial conflicts of interest related to this work.

## 2. Preliminary

### 2.1. Notation

We denote as $\|x\|_2 = \sqrt{\sum_{i=1}^d x_i^2}$ the Euclidean norm of $x \in \mathbb{R}^d$, and as $\|A\|_2 = \sqrt{\lambda_{\max}(A^\top A)}$ the spectral norm of $A \in \mathbb{R}^{r \times s}$. We denote by $\mathcal{P}(\Omega)$ the space of probability measures on $\Omega \subset \mathbb{R}^d$. We denote by $W_p$ the Wasserstein distance for $1 \leq p < \infty$. For a measurable map $T : \mathbb{R}^d \to \mathbb{R}^{d'}$ and a measure $\mu \in \mathcal{P}(\Omega)$, the push-forward $T_\sharp \mu$ is given by $T_\sharp \mu(\Omega') = \mu(T^{-1}(\Omega'))$ for a Borel set $\Omega' \subset \mathbb{R}^{d'}$. For metric spaces $(X, d_x)$ and $(Y, d_y)$, and $C > 0$, we write $f : (X, d_x) \to (Y, d_y)$ is $C$-Lipschitz continuous if it holds that
$$d_y(f(x_1), f(x_2)) \leq C d_x(x_1, x_2)$$
for all $x_1, x_2 \in X$. When the metrics $d_x$ and $d_y$ are clear from the context, we write that $f : X \to Y$ is $C$-Lipschitz continuous, omitting $d_x$ and $d_y$.

### 2.2. Lipschitz MLP layer

We define the gradient-decent-type MLP mapping $F_\xi : \mathbb{R}^d \to \mathbb{R}^d$ by
$$F_\xi(x) := x - \tau W^\top \sigma(Wx + b) \tag{1}$$
where the learnable parameter is denoted by $\xi = (W, b, \tau)$, for a choice of $W \in \mathbb{R}^{k \times d}$, $b \in \mathbb{R}^k$, and $\tau \geq 0$. The map $F_\xi$ can be interpreted as a single explicit Euler step of size $\tau$ for the negative gradient flow equation $\dot{x} = -\nabla g(x)$ where $g(x) := 1^\top \gamma(Wx + b)$, $\gamma' = \sigma$ and $1 \in \mathbb{R}^k$ is a vector of ones. ResNets with layers $F_\xi$ have been used in (Meunier et al., 2021; Prach et al., 2023; Sherry et al.,

2024) to improve the robustness to adversarial attacks, and in (Sherry et al., 2024) to approximate the proximal operator and develop a provably convergent Plug-and-Play algorithm for inverse problems. In what follows, we fix $\sigma = \mathrm{ReLU}$. We recall the following lemma.

**Lemma 1.** *(Sherry et al., 2024, Theorem 2.3 and Lemma 2.5) Assuming that $\tau \in [0, 2/\|W\|_2^2]$, then the map $F_\xi : (\mathbb{R}^d, \|\cdot\|_2) \to (\mathbb{R}^d, \|\cdot\|_2)$ defined in (1) with $\sigma = \mathrm{ReLU}$ is 1-Lipschitz continuous.*

ResNets with these types of layers were shown to be universal for the set of scalar 1-Lipschitz functions in (Murari et al., 2025) and to provide a good trade-off between robustness and computational cost compared with alternatives in (Prach et al., 2023). The networks we consider in this paper rely on $F_\xi$ as the context-free maps. We then build on the construction behind their expression to define the contextual mapping introduced in the next section.

## 3. Lipschitz gradient-decent-type Transformer

### 3.1. Lipschitz attention layer

We first recall that the standard in-context attention mapping (with a single head) is defined as follows
$$(\mu, x) \mapsto \Gamma_\theta(\mu, x) := \int \frac{e^{\langle Qx, Ky \rangle}}{\int e^{\langle Qx, Kz \rangle} d\mu(z)} Vy \, d\mu(y) \in \mathbb{R}^{d'}$$
where $\theta = (Q, K, V)$, $(\mu, x) \in \mathcal{P}(\mathbb{R}^d) \times \mathbb{R}^d$, and $Q, K, V$ are matrices called query, key, and value, respectively. This map is regarded as a mapping between vectors, depending on a probability measure $\mu$ given as input, which is why $\Gamma_\theta$ is called in-context mapping. The standard discrete attention corresponds to the case where $\mu$ is the discrete empirical measure $\mu = \frac{1}{n} \sum_{i=1}^n \delta_{x_i}$, for a choice of tokens $x_1, \cdots, x_n$. The merit of abstracting from an empirical measure to a generic measure is that we do not need to specify the number of tokens, and hence our analysis can handle an infinite number of tokens. For further details on the measure-extension of in-context attention mapping, see e.g., (Castin et al., 2024; Furuya et al., 2025). We introduce a gradient-decent-type in-context attention mapping $\Gamma_\theta : \mathcal{P}(\mathbb{R}^d) \times \mathbb{R}^d \to \mathbb{R}^d$ defined as
$$\Gamma_\theta(\mu, x) := x - \eta \int \frac{e^{\langle x, Ay \rangle}}{\int e^{\langle x, Az \rangle} d\mu(z)} Ay \, d\mu(y). \tag{2}$$
where $\eta \geq 0$, $A \in \mathbb{R}^{d \times d}$ and $\theta = (A, \eta)$. Note that the matrix $A$ corresponds to $Q^\top K$ in the standard definition, and that we fix $V = A$. The constraint $V = A$ is introduced to enable a gradient-flow interpretation of the attention layer, which plays an essential role in the proof of Lemma 2 (1). For a fixed measure $\mu$, we sometimes use the notation $\Gamma_\theta(\mu)$ for the map $\Gamma_\theta(\mu) : \mathbb{R}^d \to \mathbb{R}^d$ defined as $\Gamma_\theta(\mu)(x) =$

$\Gamma_\theta(\mu, x)$. Analogously to the MLP $F_\xi$, if $\mu \in \mathcal{P}(\Omega)$ for a compact set $\Omega \subset \mathbb{R}^d$, the map $\Gamma_\theta(\mu)$ can be interpreted as a single explicit Euler step of size $\eta$ for the negative gradient flow equation

$$\dot{x} = -\nabla\lambda(\mu)(x)$$

where $\nabla$ refers to the gradient with respect to $x$ and

$$\lambda(\mu)(x) := \log\left(\int e^{\langle x, Ay\rangle} d\mu(y)\right),$$

which is the cumulant-generating function associated with the random variable $Ay$ under $y \sim \mu$. See (Sander et al., 2022) for a cumulant-generating-function formulation of contextualizing gradient flows. From now on, with $\Gamma_\theta$ we refer to (2). We remark that under the compactness of the support set $\Omega$, $\lambda(\mu)$ is a convex and smooth function.

We show the following lemma.

**Lemma 2.** *Let $\Omega \subset \mathbb{R}^d$ be a compact set. We have the following:*

(1) *$\Gamma_\theta(\mu, \cdot)$ : $(\Omega, \|\cdot\|_2)$ $\rightarrow$ $(\mathbb{R}^d, \|\cdot\|_2)$ is 1-Lipschitz continuous for each $\mu \in \mathcal{P}(\Omega)$ for $\eta \in [0, 2/\sup_{y\in\Omega} \|Ay\|_2^2]$.*

(2) *$\Gamma_\theta(\cdot, x) : (\mathcal{P}(\Omega), W_1) \rightarrow (\mathbb{R}^d, \|\cdot\|_2)$ is $C_1$-Lipschitz continuous for each $x \in \Omega$, where $C_1 > 0$ is some constant depending on $\Omega$.*

See Appendix A for the proof. Note that, unlike gradient-descent-type models that require vanishing step sizes, we do not require the step size to go to zero in the approximation setting considered in this paper.

**Remark 3.** *Here, and in what follows, we assume that $\sup_{y\in\Omega} \|Ay\|_2^2 > 0$ since, otherwise, for every $\mu \in \mathcal{P}(\Omega)$ we have $\Gamma_\theta(\mu) = \mathrm{id}$ on $\Omega$, and all the claims on Lipschitz regularity follow trivially.*

### 3.2. Lipschitz deep Transformer

The aim of this section is to introduce a deep Transformer preserving the Lipschitz continuity. Such a Transformer follows from the composition of gradient-decent-type attentions and MLPs defined in the previous section, where each map is Lipschitz continuous. Note that, in the gradient-decent-type architecture, whereas the MLP layer is globally Lipschitz continuous, the attention layer is locally Lipschitz continuous, implying that some care is needed to properly compose these maps.

Let $\Omega, \tilde{\Omega} \subset \mathbb{R}^d$ be compact sets. We define the class of

1-Lipschitz MLP layers as

$$\begin{aligned}
\mathcal{E}_{d,\Omega,\tilde{\Omega}} := \Big\{ &F_\xi : \mathbb{R}^d \to \mathbb{R}^d : \\
&F_\xi(x) = x - \tau W^\top \sigma(Wx + b), \ F_\xi(\Omega) \subset \tilde{\Omega}, \\
&W \in \mathbb{R}^{k\times d}, \ b \in \mathbb{R}^k, \ \tau \in [0, 2/\|W\|_2^2], \ k \in \mathbb{N} \Big\}.
\end{aligned}$$

Similarly, we define the class of Lipschitz self-attention layers by

$$\begin{aligned}
\mathcal{V}_{d,\Omega,\tilde{\Omega}} := \Big\{ &\Gamma_\theta : \mathcal{P}(\Omega) \times \Omega \to \mathbb{R}^d : \\
&\Gamma_\theta(\mu, x) = x - \eta \int \frac{e^{\langle x, Ay\rangle}}{\int e^{\langle x, Az\rangle} d\mu(z)} Ay d\mu(y), \\
&\Gamma_\theta(\mathcal{P}(\Omega) \times \Omega) \subset \tilde{\Omega}, \ A \in \mathbb{R}^{d\times d}, \ \eta \in [0, 2/\sup_{y\in\Omega} \|Ay\|_2^2] \Big\}.
\end{aligned}$$

We clearly write the dependency of input domain $\Omega$ and output domain $\tilde{\Omega}$ because, as shown in Lemma 2, the attention layer $\Gamma_\theta(\mu)$ is 1-Lipschitz for an interval of admissible $\eta$ which depends on the input domain. That means that when we define a deep architectures, each layer should have a different interval of admissible $\eta$.

**Definition 4** (Propagation of the measure of tokens). Let $F_\xi \in \mathcal{E}_{d,\Omega,\tilde{\Omega}}$ and $\Gamma_\theta \in \mathcal{V}_{d,\Omega,\tilde{\Omega}}$. We define by $\bar{F}_\xi$ and $\bar{\Gamma}_\theta$ the extensions of $F_\xi$ and $\Gamma_\theta$, respectively, to handle input measures. Their definitions are as follows

$$\bar{F}_\xi : \mathcal{P}(\mathbb{R}^d) \times \mathbb{R}^d \to \mathcal{P}(\mathbb{R}^d) \times \mathbb{R}^d$$
$$\bar{F}_\xi(\mu, x) := ((F_\xi)_\sharp \mu, F_\xi(x)), \ \text{(context-free mapping)},$$

and

$$\bar{\Gamma}_\theta : \mathcal{P}(\Omega) \times \Omega \to \mathcal{P}(\mathbb{R}^d) \times \mathbb{R}^d$$
$$\bar{\Gamma}_\theta(\mu, x) := (\Gamma_\theta(\mu)_\sharp \mu, \Gamma_\theta(\mu, x)), \ \text{(in-context mapping)}.$$

We remark that the MLP $F_\xi$ transforms the measure via the pushforward map which is independent of the input measure, hence why it is called context-free, whereas $\Gamma_\theta$ takes into account the input measure, and is therefore referred to as an in-context mapping.

**Definition 5** (Lipschitz deep Transformer). Let $\Omega_1 = \Omega$. For $\ell = 1, \ldots, L$, let $\Gamma_{\theta_\ell} \in \mathcal{V}_{d,\Omega_\ell,\tilde{\Omega}_\ell}$ and $F_{\xi_\ell} \in \mathcal{E}_{d,\tilde{\Omega}_\ell,\Omega_{\ell+1}}$, where the compact sets $\tilde{\Omega}_\ell$ and $\Omega_{\ell+1}$ are defined recursively as the images of $\Omega_\ell$ under the corresponding layers, i.e., they must satisfy that for $\ell = 1, \ldots, L$

$$\Gamma_{\theta_\ell}(\mathcal{P}(\Omega_\ell) \times \Omega_\ell) \subset \tilde{\Omega}_\ell, \quad F_{\xi_\ell}(\tilde{\Omega}_\ell) \subset \Omega_{\ell+1}. \quad (3)$$

We define the deep Lipschitz Transformer by

$$T(\mu, x) := \pi_2 \circ \bar{F}_{\xi_L} \circ \bar{\Gamma}_{\theta_L} \circ \cdots \circ \bar{F}_{\xi_1} \circ \bar{\Gamma}_{\theta_1}(\mu, x), \quad (4)$$

where $\pi_2(\mu, x) := x$ denotes the projection onto the query variable.

The corresponding hypothesis class is given by

$$\mathcal{T}_{d,\Omega} := \Big\{ T : \mathcal{P}(\Omega) \times \Omega \to \mathbb{R}^d \ : \ T \text{ is of the form (4)},$$
$$L \in \mathbb{N}, \ \Gamma_{\theta_\ell} \in \mathcal{V}_{d,\Omega_\ell,\tilde{\Omega}_\ell}, \ F_{\xi_\ell} \in \mathcal{E}_{d,\tilde{\Omega}_\ell,\Omega_{\ell+1}},$$
$$\Omega_\ell, \tilde{\Omega}_\ell \subset \mathbb{R}^d \text{ compact with (3), } \ell = 1, ..., L \Big\}.$$

By Lemmata 1 and 2, we obtain the following result.

**Lemma 6.** *For any $T \in \mathcal{T}_{d,\Omega}$, the following properties hold:*

(1) *For each $\mu \in \mathcal{P}(\Omega)$, the map $T(\mu, \cdot) : (\Omega, \| \cdot \|_2) \to (\mathbb{R}^d, \| \cdot \|_2)$ is 1-Lipschitz.*

(2) *For each $x \in \Omega$, the map $T(\cdot, x) : (\mathcal{P}(\Omega), W_1) \to (\mathbb{R}^d, \| \cdot \|_2)$ is $C_2$-Lipschitz, where $C_2 > 0$ is some constant depending on $\Omega$ and $L$.*

## 4. Approximation result

### 4.1. Main result

Let $C > 0$ and $d \in \mathbb{N}$, and let $\Omega \subset \mathbb{R}^d$ be a compact set. We define the input space as

$$\mathcal{X} := \mathcal{P}(\Omega) \times \Omega.$$

We denote by $\mathcal{C}_{1,C}(\mathcal{X}, \mathbb{R})$ the class of scalar target mappings

$$\mathcal{C}_{1,C}(\mathcal{X}, \mathbb{R}) := \Big\{ F : \mathcal{X} \to \mathbb{R} :$$
$$F(\mu, \cdot) : (\Omega, \| \cdot \|_2) \to (\mathbb{R}, | \cdot |) \text{ is 1-Lipschitz}$$
$$F(\cdot, x) : (\mathcal{P}(\Omega), W_1) \to (\mathbb{R}, | \cdot |) \text{ is } C\text{-Lipschitz} \Big\}.$$

**Remark 7.** *We remark that all the functions in $\mathcal{C}_{1,C}(\mathcal{X}, \mathbb{R})$ are Lipschitz continuous in the product metric since for every pair $(\mu, x), (\nu, y) \in \mathcal{X}$ it holds*

$$|F(\nu, y) - F(\mu, x)|$$
$$= |F(\nu, y) - F(\nu, x) + F(\nu, x) - F(\mu, x)|$$
$$\leq \|y - x\|_2 + C W_1(\mu, \nu)$$
$$\leq \max\{1, C\} d((x, \mu), (y, \nu)),$$

*where $d$ is the product metric defined over $\mathcal{X} = \mathcal{P}(\Omega) \times \Omega$, i.e. $d((x, \mu), (y, \mu)) = \|y - x\|_2 + W_1(\mu, \nu)$.*

We define the class of scalar Lipschitz deep Transformers with lifting and projection:

$$\mathcal{G}_C(\mathcal{X}, \mathbb{R}) := \mathcal{C}_{1,C}(\mathcal{X}, \mathbb{R}) \cap \mathcal{K}(\mathcal{X}, \mathbb{R})$$

where

$$\mathcal{K}(\mathcal{X}, \mathbb{R}) := \Big\{ v^\top \circ T \circ \bar{Q} : \mathcal{X} \to \mathbb{R} \ :$$
$$Q \in \mathcal{Q}_{d,h,\Omega,\Omega_1}, v \in \mathbb{R}^h, T \in \mathcal{T}_{h,\Omega_1}, h \in \mathbb{N},$$
$$\Omega_1 \subset \mathbb{R}^h \text{ compact with } Q(\Omega) \subset \Omega_1 \Big\}.$$

Here, $v^\top$ denotes the linear projection onto a one-dimensional output. The map $Q$ is a lifting mapping, defined as an affine embedding

$$\mathcal{Q}_{d,h,\Omega,\Omega_1} := \{ Q : \mathbb{R}^d \to \mathbb{R}^h \ : \ Q(x) = Ax + b,$$
$$A \in \mathbb{R}^{h \times d}, b \in \mathbb{R}^h, Q(\Omega) \subset \Omega_1 \}.$$

The map $\bar{Q}$ is defined by

$$\bar{Q} : \mathcal{P}(\mathbb{R}^d) \times \mathbb{R}^d \to \mathcal{P}(\mathbb{R}^h) \times \mathbb{R}^h$$
$$\bar{Q}(\mu, x) := (Q_\sharp \mu, Q(x)), \text{ (context-free mapping)}.$$

Note that, in general, the map $v^\top \circ T \circ \bar{Q} \in \mathcal{K}(\mathcal{X}, \mathbb{R})$ does not necessarily belong to $\mathcal{C}_{1,C}(\mathcal{X}, \mathbb{R})$, since we do not impose $\|v\|_2 \leq 1$ and $\|Q\|_2 \leq 1$. This is why $\mathcal{G}_C(\mathcal{X}, \mathbb{R})$ is defined by intersecting $\mathcal{K}(\mathcal{X}, \mathbb{R})$ with $\mathcal{C}_{1,C}(\mathcal{X}, \mathbb{R})$, so we can ensure a controlled Lipschitz constant. We remark that this technique of intersecting with the target set is common in theoretical studies, see, for example, (Murari et al., 2025; Eckstein, 2020). Moreover, for any $T \in \mathcal{T}_{h,\Omega_1}$, the map $T$ is always 1-Lipschitz with respect to the query variable and $C_2$-Lipschitz with respect to the context measure, for worst-case Lipschitz constant $C_2 > 0$ given by Lemma 6.

We now ask whether a deep Transformer that is 1-Lipschitz in the query variable and $C$-Lipschitz in the context variable, for an arbitrary prescribed constant $C > 0$, independent of accuracy, can approximate any continuous in-context map with the same Lipschitz constraints. The main result of this paper answers this question in the affirmative.

**Theorem 8.** *Let $C > 0$ and $d \in \mathbb{N}$. Let $\Omega \subset \mathbb{R}^d$ be a compact set. Then, for any $\varepsilon \in (0, 1)$ and any $\Lambda^* \in \mathcal{C}_{1,C}(\mathcal{X}, \mathbb{R})$, there is $\Lambda \in \mathcal{G}_C(\mathcal{X}, \mathbb{R})$ such that*

$$\sup_{(\mu,x) \in \mathcal{X}} |\Lambda(\mu, x) - \Lambda^*(\mu, x)| \leq \varepsilon.$$

This is, to the best of our knowledge, the first theoretical approximation result for Transformer architectures that preserve Lipschitz continuity. More precisely, we show that gradient-descent-type in-context Transformers can approximate arbitrary Lipschitz continuous in-context maps.

Our result can be viewed as a generalization of (Murari et al., 2025, Theorem 3.1) to in-context mappings by incorporating an additional argument corresponding to the token measure. This extension is highly nontrivial: the proof requires a variant of the Restricted Stone–Weierstrass theorem (Lemma 9) adapted to functions of two variables, including probability measures.

Key technical ingredients used to verify the hypotheses of this theorem include concatenation arguments (Lemma 11) and Kantorovich–Rubinstein duality-based lower-bounds (Lemma 13). These tools are specific to the measure-theoretic in-context setting and are fundamentally different

from the Euclidean-space arguments employed in (Murari et al., 2025).

## 4.2. Proof of Theorem 8

The proof of Theorem 8 is based on a variant of the Restricted Stone–Weierstrass theorem, adapted to Lipschitz functions of two variables. We first state the abstract approximation result.

**Lemma 9** (Variant of the Restricted Stone–Weierstrass theorem). *Let $(U, d_U)$ and $(X, d_X)$ be compact metric spaces with at least two points, and let $C > 0$. We denote by $\mathcal{C}_{1,C}(U \times X, \mathbb{R})$ the space of all functions $f : U \times X \to \mathbb{R}$ such that $x \mapsto f(u, x)$ is 1-Lipschitz continuous for every $u \in U$, and $u \mapsto f(u, x)$ is C-Lipschitz continuous for every $x \in X$.*

*Let $L \subset \mathcal{C}_{1,C}(U \times X, \mathbb{R})$. Assume that:*

(A) *$L$ is a lattice, i.e., for any $\Lambda, \Lambda' \in L$, it holds that*

$$\max\{\Lambda, \Lambda'\} \in L, \quad \min\{\Lambda, \Lambda'\} \in L.$$

(B) *$L$ separates points in the following Lipschitz sense: for any two distinct points $(u, x), (v, y) \in U \times X$ and any $a, b \in \mathbb{R}$ satisfying*

$$|a - b| < C \, d_U(u, v) + d_X(x, y),$$

*there exists $f \in L$ such that*

$$f(u, x) = a, \quad f(v, y) = b.$$

*Then $L$ is dense in $\mathcal{C}_{1,C}(U \times X, \mathbb{R})$ with respect to the uniform norm.*

See Appendix B for the proof.

Lemma 9 is a modification of the Restricted Stone–Weierstrass theorem given by (Anil et al., 2019, Lemma 1), extended to functions of two arguments and formulated with a strict separation condition. A key feature in the proof of the Restricted Stone–Weierstrass theorem is its practical nature. Rather than solely providing an existence density result, it yields an approximation based on the lattice structure generated by minimax compositions of elementary building blocks. Condition (A) guarantees closure under these lattice operations, while condition (B) ensures the existence of elementary building blocks.

The core of the proof of Theorem 8 is therefore to verify conditions (A) and (B) for the class $\mathcal{G}_C(\mathcal{X}, \mathbb{R})$ of gradient-descent-type in-context Transformers.

**Condition (A)**    We begin by verifying the lattice property.

**Lemma 10.** *The set $\mathcal{G}_C(\mathcal{X}, \mathbb{R})$ is a lattice, i.e., for any $\Lambda, \Lambda' \in \mathcal{G}_C(\mathcal{X}, \mathbb{R})$, it holds that*

$$\max\{\Lambda, \Lambda'\} \in \mathcal{G}_C(\mathcal{X}, \mathbb{R}), \quad \min\{\Lambda, \Lambda'\} \in \mathcal{G}_C(\mathcal{X}, \mathbb{R}).$$

See Appendix C for the detailed proof.

The key idea is to show that the concatenation of two networks $\Lambda$ and $\Lambda'$ can be represented within the same architectural class, by composing MLPs and attention layers in parallel. More precisely, we observe that

$$\begin{pmatrix} \Lambda(\mu, x) \\ \Lambda'(\mu, x) \end{pmatrix} = \begin{pmatrix} v^\top & 0 \\ 0 & v'^\top \end{pmatrix} \circ \begin{pmatrix} F_{\xi_L} \\ F_{\xi'_L} \end{pmatrix} \circ \begin{pmatrix} \Gamma_{\theta_L}(\mu_L) \\ \Gamma_{\theta'_L}(\mu'_L) \end{pmatrix} \circ \cdots$$

$$\circ \begin{pmatrix} F_{\xi_1} \\ F_{\xi'_1} \end{pmatrix} \circ \begin{pmatrix} \Gamma_{\theta_1}(\mu_1) \\ \Gamma_{\theta'_1}(\mu'_1) \end{pmatrix} \circ \begin{pmatrix} Q \\ Q' \end{pmatrix}(x),$$

where $\Gamma_{\theta_\ell} \in \mathcal{V}_{d, \Omega_\ell, \tilde{\Omega}_\ell}$, $\Gamma_{\theta'_\ell} \in \mathcal{V}_{d, \Omega'_\ell, \tilde{\Omega}'_\ell}$, $F_{\xi_\ell} \in \mathcal{E}_{d', \tilde{\Omega}_\ell, \Omega_{\ell+1}}$, and $F_{\xi'_\ell} \in \mathcal{E}_{d', \tilde{\Omega}'_\ell, \Omega'_{\ell+1}}$. Using the argument of (Murari et al., 2025, Lemma 3.2), we can merge the parallel MLP layers, i.e.,

$$\begin{pmatrix} F_{\xi_\ell} \\ F_{\xi'_\ell} \end{pmatrix} = F_{\xi''_\ell}$$

for some $F_{\xi''_\ell} \in \mathcal{E}_{d+d', \tilde{\Omega}_\ell \times \tilde{\Omega}'_\ell, \Omega_{\ell+1} \times \Omega'_{\ell+1}}$.

The technical difficulty in the proof of Lemma 10 is to show that an analogous concatenation is possible for attention layers. In general, this cannot be achieved by a single attention layer. Instead, we represent the parallel of two attention layers as a composition of two attention layers. More precisely, we prove the following lemma.

**Lemma 11.** *Let $\Gamma_\theta \in \mathcal{V}_{h, \Omega, \tilde{\Omega}}$ and $\Gamma_{\theta'} \in \mathcal{V}_{h', \Omega', \tilde{\Omega}'}$. Then there exist $\Gamma_{\theta''_1} \in \mathcal{V}_{h+h', \Omega \times \Omega', \tilde{\Omega} \times \Omega'}$ and $\Gamma_{\theta''_2} \in \mathcal{V}_{h+h', \tilde{\Omega} \times \Omega', \tilde{\Omega} \times \tilde{\Omega}'}$ such that, for $(\mu, x) \in \mathcal{P}(\Omega) \times \Omega$, and $(\mu', x') \in \mathcal{P}(\Omega') \times \Omega'$, and $\gamma \in \Pi(\mu, \mu')$, where $\Pi(\mu, \mu')$ denotes the set of all couplings of $\mu$ and $\mu'$, we have*

$$\begin{pmatrix} \Gamma_\theta(\mu, x) \\ \Gamma_{\theta'}(\mu', x') \end{pmatrix} = \pi_2 \circ \bar{\Gamma}_{\theta''_2} \circ \bar{\Gamma}_{\theta''_1}\left(\gamma, \begin{pmatrix} x \\ x' \end{pmatrix}\right).$$

Here, $\pi_2$ denotes the projection onto the query variable, and should not be confused with the projection onto the second component of a Cartesian product. Note that ReLU MLPs can represent the identity map, so the constructions in Lemma 11 can be realized by the alternating composition of context-free and in-context maps. See Appendix D for the proof.

**Condition (B)**    We next verify the separation property.

**Lemma 12.** *Let $(\mu, x), (\mu', x') \in \mathcal{X}$ with $(\mu, x) \neq (\mu', x')$, and let $a, b \in \mathbb{R}$ with $a \neq b$, satisfy*

$$|a - b| < \|x - x'\|_2 + C \, W_1(\mu, \mu').$$

*Then there exists $\Lambda \in \mathcal{G}_C(\mathcal{X}, \mathbb{R})$ such that*

$$\Lambda(\mu, x) = a, \qquad \Lambda(\mu', x') = b.$$

See Appendix E for the detailed proof.

The proof of Lemma 12 relies on the following lower-bound result.

**Lemma 13.** *Let* $(\mu, x), (\mu', x') \in \mathcal{X}$. *Then for any* $\varepsilon \in (0, 1)$, *there exists* $\Lambda \in \mathcal{G}_C(\mathcal{X})$ *such that*

$$C\big(W_1(\mu, \mu') - \varepsilon\big) \le \Lambda(\mu, x) - \Lambda(\mu', x').$$

We briefly outline the main idea of the proof. Let

$$\mathcal{F} := \big\{\varphi : \Omega \to \mathbb{R} \mid \mathrm{Lip}(\varphi) \le 1, \ \varphi(x_0) = 0\big\},$$

for some fixed $x_0 \in \Omega$. By the Arzelà–Ascoli theorem, $\mathcal{F}$ is compact. Since the map $\varphi \mapsto \int \varphi \, d(\mu - \mu')$ is continuous, there exists $\varphi_0 \in \mathcal{F}$ such that

$$W_1(\mu, \mu') = \sup_{\varphi \in \mathcal{F}} \int \varphi \, d(\mu - \mu') = \int \varphi_0 \, d(\mu - \mu').$$

By (Murari et al., 2025, Theorem 3.1), there exist a compact set $\Omega_1 \subset \mathbb{R}^h$, $Q \in \mathcal{Q}_{d,h,\Omega,\Omega_1}$, $v \in \mathbb{R}^h$, and a 1-Lipschitz map $F : \mathbb{R}^h \to \mathbb{R}^h$ given as a composition of gradient-decent-type MLPs such that

$$\sup_{z \in \Omega} \big|\varphi_0(z) - v^\top \circ F \circ Q(z)\big| \le \varepsilon/2.$$

This implies

$$\begin{aligned} W_1(\mu, \mu') &= \int \varphi_0 \, d(\mu - \mu') \\ &\le \int v^\top \circ F \circ Q \, d(\mu - \mu') + \varepsilon. \end{aligned}$$

Finally, the term $\int v^\top \circ F \circ Q \, d(\mu - \mu')$ can be written as

$$\Lambda(\mu, x) - \Lambda(\mu', x')$$

for a suitable $\Lambda \in \mathcal{G}_C(\mathcal{X}, \mathbb{R})$. We remark that composing context-free or in-context maps successively is possible under our architectural assumptions, since both context-free and in-context maps in $\mathcal{G}_C(\mathcal{X}, \mathbb{R})$ can represent the identity map. Therefore, composing multiple context-free MLPs to realize $F$ does not violate the in-context formulation. Moreover, the integration operation in $\int v^\top \circ F \circ Q \, d(\mu - \mu')$ can be implemented by a single in-context layer. See Appendix F for the complete proof.

**Remark 14.** *We remark that the gradient-descent-type in-context Transformers constructed in the proof are not fully constructive, since the construction relies on existence results such as the Arzelà–Ascoli theorem to obtain an optimizer $\varphi_0 \in \mathcal{F}$ in the Kantorovich–Rubinstein dual formulation. However, apart from the non-explicit realization of $\varphi_0 \in \mathcal{F}$, the remaining argument is constructive, since the Restricted Stone–Weierstrass step yields explicit approximants via minimax compositions.*

## 5. Discussion

We introduced a new gradient-descent type in-context mapping. This consists of an attention mechanism interpreted as a nonexpansive update of the query driven by a context-dependent potential. We then studied its approximation properties in an abstract setting where the context is modeled as a probability measure, and its Lipschitz continuity with respect to the context is controlled in the Wasserstein-1 metric. In particular, we established a universal approximation theorem for Lipschitz-continuous in-context Transformers built from these Lipschitz primitives. Beyond existence, the proof is inherently constructive: a Restricted Stone-Weierstrass argument reduces density to the ability to build minimax compositions of elementary Lipschitz primitives, and, in the measure-theoretic setting, these primitives arise naturally from Kantorovich-Rubinstein duality, although the resulting Lipschitz primitives are not given in an explicit form. This perspective yields concrete guidance for the design of Lipschitz-preserving in-context architectures and highlights two structural properties that drive universality in our approach: closure under lattice-type operations (Lemma 11) and the ability to represent the identity map.

Several directions are required to bridge the gap between this abstract universality result and practical, certifiable architectures. First, the step size $\eta_\ell$ in our in-context attention layers depends on the layerwise input domain $\Omega_\ell$. This input domain is, in general, difficult to estimate precisely. Second, while context-free Lipschitz control can be enforced by spectral-norm constraints, the corresponding bound for in-context attention is expressed through a supremum quantity that may be difficult to estimate or certify in practice. A natural next step is therefore to design alternative Lipschitz in-context mappings whose Lipschitz constants admit explicit, tractable upper bounds while preserving the algebraic properties used in the density proof. Since our argument is modular, we expect the same proof strategy to transfer to other in-context mechanisms that retain (i) lattice-type closure and (ii) identity representability.

Second, our result is an existence theorem based on a Restricted Stone–Weierstrass type argument, which is inherently non-quantitative. Deriving explicit bounds on the lifting dimension or the network size and depth in terms of $\varepsilon$ would require different techniques, and we leave this for future work. Such quantitative estimates would also strengthen the connection to statistical learning analysis.

Finally, our analysis focuses on scalar-valued targets. Although extending universality to vector-valued outputs equipped with the $\ell^\infty$ norm is straightforward by a coordinate-wise argument and taking the maximum over coordinates, extending universality to vector-valued outputs equipped with the $\ell^2$ norm is nontrivial. This difficulty already appears in the context-free 1-Lipschitz setting and will

likely require new approximation mechanisms beyond the scalar proof strategy. A key obstruction is that the pointwise lattice structure (closure under max and min) that underpins the Restricted Stone-Weierstrass argument in the scalar case does not extend canonically to vector-valued mappings, where no natural total order is available. Establishing such extensions would enable a more comprehensive approximation theory for Lipschitz Transformers and further clarify which Lipschitz-preserving design choices are most compatible with practice.

## Impact Statement

This paper advances the theoretical understanding of Lipschitz-continuous Transformer architectures by establishing approximation results in an in-context setting. The contributions are foundational and do not target a specific application domain. Any downstream societal impacts depend on how such architectures are deployed in practice and are therefore subject to the usual considerations associated with machine learning models.

## Acknowledgements

TF is supported by JSPS KAKENHI Grant Number JP24K16949, 25H01453, JST CREST JPMJCR24Q5, JST ASPIRE JPMJAP2329. DM acknowledges support from the EPSRC programme grant in 'The Mathematics of Deep Learning', under the project EP/V026259/1. CBS acknowledges support from the Philip Leverhulme Prize, the Royal Society Wolfson Fellowship, the EPSRC advanced career fellowship EP/V029428/1, the EPSRC programme grant EP/V026259/1, and the EPSRC grants EP/S026045/1 and EP/T003553/1, EP/N014588/1, EP/T017961/1, the Wellcome Innovator Awards 215733/Z/19/Z and 221633/Z/20/Z, the European Union Horizon 2020 research and innovation programme under the Marie Skodowska-Curie grant agreement NoMADS and REMODEL, the Cantab Capital Institute for the Mathematics of Information and the Alan Turing Institute. This research was also supported by the NIHR Cambridge Biomedical Research Centre (NIHR203312). The views expressed are those of the author(s) and not necessarily those of the NIHR or the Department of Health and Social Care.

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

# A. Proof of Lemma 2

(1) Fix $\mu \in \mathcal{P}(\Omega)$. Recall that $\Gamma_\theta(\mu, x)$ is given by

$$\Gamma_\theta(\mu, x) = x - \eta \int \frac{e^{\langle x, Ay \rangle}}{\int e^{\langle x, Az \rangle} \, d\mu(z)} Ay \, d\mu(y).$$

We compute the Jacobian of the second term with respect to $x$. A direct calculation yields

$$\begin{aligned}
\nabla_x \int \frac{e^{\langle x, Ay \rangle}}{\int e^{\langle x, Az \rangle} \, d\mu(z)} Ay \, d\mu(y) &= \int \frac{e^{\langle x, Ay \rangle}}{\int e^{\langle x, Az \rangle} \, d\mu(z)} (Ay - m(x))(Ay)^\top \, d\mu(y) \\
&= \mathbb{E}_{\mu_x} \big[ (AY - m(x))(AY)^\top \big] \\
&= \mathbb{E}_{\mu_x} \big[ (AY - m(x))(AY - m(x))^\top \big] \\
&= \mathrm{Cov}_{\mu_x} [AY],
\end{aligned}$$

where

$$m(x) := \int \frac{e^{\langle x, Ay \rangle}}{\int e^{\langle x, Az \rangle} \, d\mu(z)} Ay \, d\mu(y) = \mathbb{E}_{\mu_x}[AY],$$

and $\mu_x$ denotes the probability measure

$$d\mu_x(y) = \frac{e^{\langle x, Ay \rangle}}{\int e^{\langle x, Az \rangle} \, d\mu(z)} \, d\mu(y).$$

Let $v \in \mathbb{R}^d$ with $\|v\|_2 = 1$. Then

$$\begin{aligned}
v^\top \mathrm{Cov}_{\mu_x}[AY] v &= \mathbb{E}_{\mu_x} \big[ (v^\top (AY - m(x)))^2 \big] \\
&\leq \mathbb{E}_{\mu_x} \big[ \|AY - m(x)\|_2^2 \big] \\
&= \mathbb{E}_{\mu_x} \big[ \|AY\|_2^2 \big] - \|m(x)\|_2^2 \\
&\leq \sup_{y \in \Omega} \|Ay\|_2^2.
\end{aligned}$$

Since this bound holds for all unit vectors $v$, we obtain

$$\left\| \nabla_x \int \frac{e^{\langle x, Ay \rangle}}{\int e^{\langle x, Az \rangle} \, d\mu(z)} Ay \, d\mu(y) \right\|_2 \leq \|\mathrm{Cov}_{\mu_x}[AY]\|_2 \leq \sup_{y \in \Omega} \|Ay\|_2^2.$$

Consequently, the mapping

$$x \longmapsto \Gamma_\theta(\mu, x) = x - \eta \int \frac{e^{\langle x, Ay \rangle}}{\int e^{\langle x, Az \rangle} \, d\mu(z)} Ay \, d\mu(y)$$

is 1-Lipschitz on $\Omega$ provided that

$$\eta \in \big[ 0, \, 2/\sup_{y \in \Omega} \|Ay\|_2^2 \big],$$

by (Sherry et al., 2024, Theorem 2.8). This completes the proof of (1).

(2) Fix $x \in \Omega$. Define

$$f_x(y) := \langle x, Ay \rangle, \qquad Z_x(\mu) := \int_\Omega e^{f_x(z)} \, d\mu(z), \qquad N_x(\mu) := \int_\Omega e^{f_x(y)} Ay \, d\mu(y).$$

Then

$$\Gamma_\theta(\mu, x) = x - \eta \frac{N_x(\mu)}{Z_x(\mu)}.$$

Since $\Omega$ is bounded, set

$$R := \sup_{y \in \Omega} \|y\|_2 < \infty.$$

Then

$$|f_x(y)| \le \|x\|_2 \|A\|_{\mathrm{op}} \|y\|_2 \le \|A\|_{\mathrm{op}} R^2 =: B,$$

so that

$$e^{-B} \le e^{f_x(y)} \le e^B, \qquad Z_x(\mu) \in [e^{-B}, e^B] \quad \text{for all } \mu \in \mathcal{P}(\Omega).$$

The function $f_x$ is Lipschitz with respect to $y$ with

$$\mathrm{Lip}(f_x) \le \|A^\top x\|_2 \le \|A\|_{\mathrm{op}} R =: L_f.$$

Using the mean value theorem and Kantorovich–Rubinstein duality,

$$|Z_x(\mu) - Z_x(\nu)| \le \mathrm{Lip}(e^{f_x}) W_1(\mu, \nu) \le e^B L_f W_1(\mu, \nu).$$

For $g(y) := e^{f_x(y)} A y$, we estimate

$$\|g(y) - g(y')\|_2 \le \|Ay\|_2 |e^{f_x(y)} - e^{f_x(y')}| + e^{f_x(y')} \|A(y - y')\|_2$$
$$\le e^B \|A\|_{\mathrm{op}} (R L_f + 1) \|y - y'\|_2.$$

Hence

$$\|N_x(\mu) - N_x(\nu)\|_2 \le e^B \|A\|_{\mathrm{op}} (1 + R L_f) W_1(\mu, \nu),$$

and moreover

$$\|N_x(\nu)\|_2 \le \int_\Omega e^{f_x(y)} \|Ay\|_2 \, d\nu(y) \le e^B \|A\|_{\mathrm{op}} R.$$

We write

$$\frac{N_x(\mu)}{Z_x(\mu)} - \frac{N_x(\nu)}{Z_x(\nu)} = \frac{N_x(\mu) - N_x(\nu)}{Z_x(\mu)} + N_x(\nu) \Big( \frac{1}{Z_x(\mu)} - \frac{1}{Z_x(\nu)} \Big).$$

Since

$$\Big| \frac{1}{Z_x(\mu)} - \frac{1}{Z_x(\nu)} \Big| \le e^{2B} |Z_x(\mu) - Z_x(\nu)|,$$

we obtain

$$\Big\| \frac{N_x(\mu)}{Z_x(\mu)} - \frac{N_x(\nu)}{Z_x(\nu)} \Big\|_2 \le C \, W_1(\mu, \nu),$$

where $C > 0$ depends only on $A, \Omega$.

Finally,

$$\|\Gamma_\theta(\mu, x) - \Gamma_\theta(\nu, x)\|_2 = \eta \Big\| \frac{N_x(\mu)}{Z_x(\mu)} - \frac{N_x(\nu)}{Z_x(\nu)} \Big\|_2 \le C_1 \, W_1(\mu, \nu),$$

which completes the proof.

$\square$

## B. Proof of Lemma 9

The proof follows from a slight modification of (Anil et al., 2019, Lemma 1), adapted to functions of two variables with possibly different Lipschitz constants.

Let $g \in \mathcal{C}_{1,C}(U \times X, \mathbb{R})$ and let $\varepsilon \in (0, 1)$. Since $U \times X$ is compact and $g$ is continuous, we have

$$\|g\|_\infty := \sup_{(u,x) \in U \times X} |g(x, u)| < \infty.$$

Define

$$g_\varepsilon := \Big( 1 - \frac{\varepsilon}{2(1 + \|g\|_\infty)} \Big) g.$$

Then
$$|g_\varepsilon(u, x) - g(u, x)| \le \varepsilon/2 \qquad \text{for all } (u, x) \in U \times X,$$

and $g_\varepsilon \in \mathcal{C}_{1,C}(U \times X, \mathbb{R})$. Thus, it suffices to approximate $g_\varepsilon$ within accuracy $\varepsilon/2$.

Fix $(u, x) \in U \times X$. For each $(v, y) \in U \times X$, by the separation property (B) of $L$, there exists a function $f_{(v,y)} \in L$ such that
$$f_{(v,y)}(u, x) = g_\varepsilon(u, x), \qquad f_{(v,y)}(v, y) = g_\varepsilon(v, y).$$

Indeed, since $x \mapsto g_\varepsilon(u, x)$ is $\left(1 - \frac{\varepsilon}{2(1+\|g\|_\infty)}\right)$-Lipschitz and $u \mapsto g_\varepsilon(u, x)$ is $C\left(1 - \frac{\varepsilon}{2(1+\|g\|_\infty)}\right)$-Lipschitz, we have

$$\begin{aligned} |g_\varepsilon(u, x) - g_\varepsilon(v, y)| &\le \left(1 - \frac{\varepsilon}{2(1+\|g\|_\infty)}\right) \left(d_X(x, y) + C\, d_U(u, v)\right) \\ &< d_X(x, y) + C\, d_U(u, v), \end{aligned}$$

which allows us to apply condition (B).

Define the open set
$$V_{(v,y)} := \left\{(w, z) \in U \times X \mid f_{(v,y)}(w, z) < g_\varepsilon(w, z) + \varepsilon/2\right\}.$$

Then $V_{(v,y)}$ is open and contains both $(u, x)$ and $(v, y)$. Hence, the family $\{V_{(v,y)}\}_{(v,y) \in U \times X}$ forms an open cover of $U \times X$. By compactness, there exists a finite subcover $\{V_{(v_1,y_1)}, \dots, V_{(v_n,y_n)}\}$.

Define
$$F_{(u,x)} := \min\{f_{(v_1,y_1)}, \dots, f_{(v_n,y_n)}\}.$$

Since $L$ is a lattice by assumption (A), we have $F_{(u,x)} \in L$. Moreover,
$$F_{(u,x)}(u, x) = g_\varepsilon(u, x), \qquad F_{(u,x)}(w, z) < g_\varepsilon(w, z) + \varepsilon/2 \quad \text{for all } (w, z) \in U \times X.$$

Next, define
$$U_{(u,x)} := \left\{(w, z) \in U \times X \mid F_{(u,x)}(w, z) > g_\varepsilon(w, z) - \varepsilon/2\right\}.$$

Then $U_{(u,x)}$ is an open neighborhood of $(u, x)$. The collection $\{U_{(u,x)}\}_{(u,x) \in U \times X}$ forms an open cover of $U \times X$, and hence admits a finite subcover $\{U_{(u_1,x_1)}, \dots, U_{(u_m,x_m)}\}$.

Define
$$G := \max\{F_{(u_1,x_1)}, \dots, F_{(u_m,x_m)}\}.$$

Again, by the lattice property, $G \in L$. Furthermore, for all $(w, z) \in U \times X$, we have
$$g_\varepsilon(w, z) - \varepsilon/2 < G(w, z) < g_\varepsilon(w, z) + \varepsilon/2.$$

Therefore,
$$\|g_\varepsilon - G\|_\infty < \varepsilon/2.$$

Thus, we conclude that $L$ is dense in $\mathcal{C}_{1,C}(U \times X, \mathbb{R})$ with respect to the uniform norm. $\qquad\square$

## C. Proof of Lemma 10

Let $\Lambda \in \mathcal{G}_C(\mathcal{X}, \mathbb{R})$. By definition, $\Lambda$ admits a representation of the form

$$\begin{aligned} \Lambda(\mu, x) &= v^\top \circ \pi_2 \circ \bar{F}_{\xi_L} \circ \bar{\Gamma}_{\theta_L} \circ \cdots \circ \bar{F}_{\xi_1} \circ \bar{\Gamma}_{\theta_1} \circ \bar{Q}(\mu, x) \\ &= v^\top \circ F_{\xi_L} \circ \Gamma_{\theta_L}(\mu_L) \circ \cdots \circ F_{\xi_1} \circ \Gamma_{\theta_1}(\mu_1) \circ Q(x), \end{aligned}$$

where the intermediate measures and features are defined recursively by

$$\begin{aligned} \mu_1 &:= Q_\sharp \mu, & z_1 &:= Q(x), \\ \mu_{\ell+1} &:= (F_{\xi_\ell} \circ \Gamma_{\theta_\ell}(\mu_\ell))_\sharp \mu_\ell, & z_{\ell+1} &:= F_{\xi_\ell} \circ \Gamma_{\theta_\ell}(\mu_\ell)(z_\ell), \end{aligned}$$

for $\ell = 1, \ldots, L - 1$.

Let $\Lambda, \Lambda' \in \mathcal{G}_C(\mathcal{X}, \mathbb{R})$ be given by

$$\Lambda(\mu, x) = v^\top \circ F_{\xi_L} \circ \Gamma_{\theta_L}(\mu_L) \circ \cdots \circ F_{\xi_1} \circ \Gamma_{\theta_1}(\mu_1) \circ Q(x),$$
$$\Lambda'(\mu, x) = v'^\top \circ F_{\xi'_L} \circ \Gamma_{\theta'_L}(\mu'_L) \circ \cdots \circ F_{\xi'_1} \circ \Gamma_{\theta'_1}(\mu'_1) \circ Q'(x),$$

with

$$\Gamma_{\theta_\ell} \in \mathcal{V}_{h, \Omega_\ell, \tilde{\Omega}_\ell}, \quad \Gamma_{\theta'_\ell} \in \mathcal{V}_{h', \Omega'_\ell, \tilde{\Omega}'_\ell}, \quad F_{\xi_\ell} \in \mathcal{E}_{h, \tilde{\Omega}_\ell, \Omega_{\ell+1}}, \quad F_{\xi'_\ell} \in \mathcal{E}_{h', \tilde{\Omega}'_\ell, \Omega'_{\ell+1}}.$$

Without loss of generality, we may assume that both networks have the same depth, since identity mappings can be represented by attention and MLP layers.

We first concatenate the two networks in parallel:

$$\begin{pmatrix} \Lambda(\mu, x) \\ \Lambda'(\mu, x) \end{pmatrix} = \begin{pmatrix} v^\top & 0 \\ 0 & v'^\top \end{pmatrix} \circ \begin{pmatrix} F_{\xi_L} \\ F_{\xi'_L} \end{pmatrix} \circ \begin{pmatrix} \Gamma_{\theta_L}(\mu_L) \\ \Gamma_{\theta'_L}(\mu'_L) \end{pmatrix} \circ \cdots \circ \begin{pmatrix} F_{\xi_1} \\ F_{\xi'_1} \end{pmatrix} \circ \begin{pmatrix} \Gamma_{\theta_1}(\mu_1) \\ \Gamma_{\theta'_1}(\mu'_1) \end{pmatrix} \circ \begin{pmatrix} Q \\ Q' \end{pmatrix}(x).$$

By (Murari et al., 2025, Lemma 3.2), each parallel MLP block can be merged, i.e.,

$$\begin{pmatrix} F_{\xi_\ell} \\ F_{\xi'_\ell} \end{pmatrix} = F_{\xi''_\ell} \quad \text{for some} \quad F_{\xi''_\ell} \in \mathcal{E}_{h+h', \tilde{\Omega}_\ell \times \tilde{\Omega}'_\ell, \Omega_{\ell+1} \times \Omega'_{\ell+1}}.$$

For the attention layers, we invoke Lemma 11, which shows that for each $\ell$,

$$\begin{pmatrix} \Gamma_{\theta_\ell}(\mu_\ell, z_\ell) \\ \Gamma_{\theta'_\ell}(\mu'_\ell, z'_\ell) \end{pmatrix} = \pi_2 \circ \bar{\Gamma}_{\theta''_{\ell,2}} \circ \bar{\Gamma}_{\theta''_{\ell,1}} \left( \gamma_\ell, \begin{pmatrix} z_\ell \\ z'_\ell \end{pmatrix} \right), \quad \gamma_\ell \in \Pi(\mu_\ell, \mu'_\ell),$$

for suitable

$$\Gamma_{\theta''_{\ell,1}} \in \mathcal{V}_{h+h', \Omega_\ell \times \Omega'_\ell, \tilde{\Omega}_\ell \times \Omega'_\ell}, \qquad \Gamma_{\theta''_{\ell,2}} \in \mathcal{V}_{h+h', \tilde{\Omega}_\ell \times \Omega'_\ell, \tilde{\Omega}_\ell \times \tilde{\Omega}'_\ell}.$$

Combining these constructions, we obtain

$$\begin{pmatrix} \Lambda(\mu, x) \\ \Lambda'(\mu, x) \end{pmatrix} = \begin{pmatrix} v^\top & 0 \\ 0 & v'^\top \end{pmatrix} \circ T'' \circ \bar{Q}''(\mu, x),$$

for some $T'' \in \mathcal{T}_{h+h', \Omega_1 \times \Omega'_1}$ and $Q'' \in \mathcal{Q}_{d, h+h', \Omega, \Omega_1 \times \Omega'_1}$.

Without loss of generality, assume $\|v\|_2 \geq \|v'\|_2$. Then

$$\begin{pmatrix} \Lambda(\mu, x)/\|v\|_2 \\ \Lambda'(\mu, x)/\|v\|_2 \end{pmatrix} = \begin{pmatrix} v^\top/\|v\|_2 & 0 \\ 0 & v'^\top/\|v\|_2 \end{pmatrix} \circ T'' \circ \bar{Q}''(\mu, x),$$

with both coefficient vectors having Euclidean norm at most 1. By the same argument as in (Murari et al., 2025, Lemma 3.2), this implies

$$\max\{\Lambda(\mu, x), \Lambda'(\mu, x)\} = (\|v\|_2 \, v'')^\top \circ F_{\xi_{L+1}} \circ T'' \circ \bar{Q}''(\mu, x),$$

for some $v'' \in \mathbb{R}^{h+h'}$ with $\|v''\|_2 \leq 1$, and $F_{\xi_{L+1}} \circ T'' \in \mathcal{T}_{h+h', \Omega_1 \times \Omega'_1}$. Thus, $(\mu, x) \mapsto \max\{\Lambda(\mu, x), \Lambda'(\mu, x)\} \in \mathcal{K}(\mathcal{X}, \mathbb{R})$.

Since pointwise maxima preserve the $(1, C)$-Lipschitz property, we conclude that

$$(\mu, x) \longmapsto \max\{\Lambda(\mu, x), \Lambda'(\mu, x)\} \in \mathcal{G}_C(\mathcal{X}, \mathbb{R}).$$

The case of the minimum follows analogously. This completes the proof. $\qquad \square$

## D. Proof of Lemma 11

Let $\Gamma_\theta \in \mathcal{V}_{h,\Omega,\tilde{\Omega}}$ and $\Gamma_{\theta'} \in \mathcal{V}_{h',\Omega',\tilde{\Omega}'}$ be given. By definition, they admit the representations

$$\Gamma_\theta(\mu, x) = x - \eta \int \frac{e^{\langle x, Ay \rangle}}{\int e^{\langle x, Az \rangle} \, d\mu(z)} Ay \, d\mu(y), \qquad A \in \mathbb{R}^{h \times h},$$

$$\Gamma_{\theta'}(\mu', x') = x' - \eta' \int \frac{e^{\langle x', A'y' \rangle}}{\int e^{\langle x', A'z' \rangle} \, d\mu'(z')} A'y' \, d\mu'(y'), \qquad A' \in \mathbb{R}^{h' \times h'},$$

with step sizes

$$\eta \in \left[0, \frac{2}{\sup_{x \in \Omega} \|Ax\|_2^2}\right], \qquad \eta' \in \left[0, \frac{2}{\sup_{x' \in \Omega'} \|A'x'\|_2^2}\right].$$

**Step 1: construction of the first attention layer.** Define $\Gamma_{\theta_1''}$ by

$$\Gamma_{\theta_1''}\left(\gamma, \begin{pmatrix} x \\ x' \end{pmatrix}\right) := \begin{pmatrix} x \\ x' \end{pmatrix} - \eta_1'' \int \frac{\exp\left(\langle \begin{pmatrix} x \\ x' \end{pmatrix}, A_1'' \begin{pmatrix} y \\ y' \end{pmatrix} \rangle\right)}{\int \exp\left(\langle \begin{pmatrix} x \\ x' \end{pmatrix}, A_1'' \begin{pmatrix} z \\ z' \end{pmatrix} \rangle\right) d\gamma(z, z')} A_1'' \begin{pmatrix} y \\ y' \end{pmatrix} d\gamma(y, y'),$$

where $\eta_1'' := \eta$ and

$$A_1'' := \begin{pmatrix} A & 0 \\ 0 & 0 \end{pmatrix}.$$

Since

$$\sup_{(x,x') \in \Omega \times \Omega'} \left\|A_1'' \begin{pmatrix} x \\ x' \end{pmatrix}\right\|_2 = \sup_{x \in \Omega} \|Ax\|_2,$$

the step-size condition for $\Gamma_{\theta_1''}$ is satisfied. Moreover, by construction,

$$\Gamma_{\theta_1''}\left(\gamma, \begin{pmatrix} x \\ x' \end{pmatrix}\right) = \begin{pmatrix} \Gamma_\theta(\mu, x) \\ x' \end{pmatrix} \in \tilde{\Omega} \times \Omega'.$$

Hence,

$$\Gamma_{\theta_1''} \in \mathcal{V}_{h+h', \Omega \times \Omega', \tilde{\Omega} \times \Omega'}.$$

We write as

$$\tilde{\gamma} := \Gamma_{\theta_1''}(\gamma)_\sharp \gamma.$$

**Step 2: construction of the second attention layer.** Define $\Gamma_{\theta_2''}$ by

$$\Gamma_{\theta_2''}\left(\tilde{\gamma}, \begin{pmatrix} \tilde{x} \\ x' \end{pmatrix}\right) := \begin{pmatrix} \tilde{x} \\ x' \end{pmatrix} - \eta_2'' \int \frac{\exp\left(\langle \begin{pmatrix} \tilde{x} \\ x' \end{pmatrix}, A_2'' \begin{pmatrix} \tilde{y} \\ y' \end{pmatrix} \rangle\right)}{\int \exp\left(\langle \begin{pmatrix} \tilde{x} \\ x' \end{pmatrix}, A_2'' \begin{pmatrix} \tilde{z} \\ z' \end{pmatrix} \rangle\right) d\tilde{\gamma}} (\tilde{z}, z') A_2'' \begin{pmatrix} \tilde{y} \\ y' \end{pmatrix} d\tilde{\gamma}(\tilde{y}, y'),$$

where $\eta_2'' := \eta'$ and

$$A_2'' := \begin{pmatrix} 0 & 0 \\ 0 & A' \end{pmatrix}.$$

As before,

$$\sup_{(\tilde{x},x') \in \tilde{\Omega} \times \Omega'} \left\|A_2'' \begin{pmatrix} \tilde{x} \\ x' \end{pmatrix}\right\|_2 = \sup_{x' \in \Omega'} \|A'x'\|_2,$$

and hence $\Gamma_{\theta_2''}$ is well-defined. Moreover,

$$\Gamma_{\theta_2''}\left(\tilde{\gamma}, \begin{pmatrix} \tilde{x} \\ x' \end{pmatrix}\right) = \begin{pmatrix} \tilde{x} \\ \Gamma_{\theta'}(\mu', x') \end{pmatrix} \in \tilde{\Omega} \times \tilde{\Omega}'.$$

Thus,

$$\Gamma_{\theta_2''} \in \mathcal{V}_{h+h', \tilde{\Omega} \times \Omega', \tilde{\Omega} \times \tilde{\Omega}'}.$$

**Step 3: composition.** By construction,

$$
\begin{aligned}
\pi_2 \circ \bar{\Gamma}_{\theta_2''} \circ \bar{\Gamma}_{\theta_1''}\left(\gamma, \begin{pmatrix} x \\ x' \end{pmatrix}\right) &= \pi_2 \circ \bar{\Gamma}_{\theta_2''}\left(\bar{\Gamma}_{\theta_1''}\left(\gamma, \begin{pmatrix} x \\ x' \end{pmatrix}\right)\right) \\
&= \pi_2 \circ \bar{\Gamma}_{\theta_2''}\left(\tilde{\gamma}, \begin{pmatrix} \Gamma_\theta(\mu, x) \\ x' \end{pmatrix}\right) \\
&= \pi_2\left(\Gamma_{\theta_2''}(\tilde{\gamma})_\sharp \tilde{\gamma}, \begin{pmatrix} \Gamma_\theta(\mu, x) \\ \Gamma_{\theta'}(\mu', x') \end{pmatrix}\right) \\
&= \begin{pmatrix} \Gamma_\theta(\mu, x) \\ \Gamma_{\theta'}(\mu', x') \end{pmatrix}.
\end{aligned}
$$

This completes the proof. $\qquad \square$

## E. Proof of Lemma 12

Let $(\mu, x) \neq (\mu', x')$ and let $a, b \in \mathbb{R}$ satisfy

$$|a - b| < \|x - x'\|_2 + C\,W_1(\mu, \mu').$$

Without loss of generality, we may assume that $a \geq b$. We fix $\varepsilon \in (0, 1)$ sufficiently small such that

$$\frac{a - b}{\|x - x'\|_2 + C\big(W_1(\mu, \mu') - \varepsilon\big)} \leq 1,$$

which is possible due to the strict inequality $|a - b| < \|x - x'\|_2 + C\,W_1(\mu, \mu')$.

By Lemma 13, there exists $\Lambda_{\mu,\mu'} \in \mathcal{G}_C(\mathcal{X})$ such that

$$C\big(W_1(\mu, \mu') - \varepsilon\big) \leq \Lambda_{\mu,\mu'}(\mu, x) - \Lambda_{\mu,\mu'}(\mu', x').$$

Moreover, from the construction (7) in the proof of Lemma 13, we may assume that $z \mapsto \Lambda_{\mu,\mu'}(\nu, z)$ is a constant function.

Since both MLPs and attention layers can represent the identity map with respect to the query variable, there exist $T \in \mathcal{T}_{d,\Omega_1}$ and $Q \in \mathcal{Q}_{d,d,\Omega,\Omega_1}$ such that

$$T \circ \bar{Q}(\nu, z) = z.$$

Define

$$
v_{x,x'} := \begin{cases} \dfrac{x - x'}{\|x - x'\|_2}, & x \neq x', \\ 0, & x = x', \end{cases}
$$

and

$$
\Lambda_{x,x'}(\nu, z) := v_{x,x'}^\top \circ T \circ \bar{Q}(\nu, z) = \begin{cases} \dfrac{1}{\|x - x'\|_2}\langle x - x', z\rangle, & x \neq x', \\ 0, & x = x'. \end{cases}
$$

Note that $\nu \mapsto \Lambda_{x,x'}(\nu, z)$ is constant, and $z \mapsto \Lambda_{x,x'}(\nu, z)$ is 1-Lipschitz.

We now define

$$\Lambda := \Lambda_{x,x'} + \Lambda_{\mu,\mu'}.$$

By construction, $\Lambda \in \mathcal{C}_{1,C}(\mathcal{X}, \mathbb{R})$. Furthermore, by the concatenation arguments used in the proofs of Lemmas 10 and 11, we conclude that $\Lambda \in \mathcal{K}(\mathcal{X}, \mathbb{R})$, and hence

$$\Lambda \in \mathcal{G}_C(\mathcal{X}, \mathbb{R}).$$

We estimate

$$|\Lambda(\mu, x) - \Lambda(\mu', x')| \geq \Lambda(\mu, x) - \Lambda(\mu', x') = \Lambda_{x,x'}(\mu, x) - \Lambda_{x,x'}(\mu', x') + \Lambda_{\mu,\mu'}(\mu, x) - \Lambda_{\mu,\mu'}(\mu', x')$$
$$\geq \|x - x'\|_2 + C\big(W_1(\mu, \mu') - \varepsilon\big). \tag{5}$$

Finally, define

$$\hat{\Lambda}(\nu, z) := b + (a - b)\, \frac{\Lambda(\nu, z) - \Lambda(\mu', x')}{\Lambda(\mu, x) - \Lambda(\mu', x')}.$$

Then

$$\hat{\Lambda}(\mu, x) = a, \qquad \hat{\Lambda}(\mu', x') = b.$$

Moreover, $\hat{\Lambda} \in \mathcal{G}_C(\mathcal{X}, \mathbb{R})$. Indeed, $\hat{\Lambda} \in \mathcal{K}(\mathcal{X}, \mathbb{R})$. Writing $\Lambda = v^\top \circ T \circ \bar{Q}$ and $\hat{\Lambda} = \alpha \Lambda + \beta$ where $\alpha \in [0, 1]$, we can rewrite

$$\hat{\Lambda} = v^\top \circ F \circ T \circ \bar{Q},$$

where $F$ is a context-free MLP implementing the affine map. Hence $\hat{\Lambda} \in \mathcal{K}(\mathcal{X}, \mathbb{R})$. Moreover, we see taht $\hat{\Lambda} \in \mathcal{C}_{1,C}(\mathcal{X}, \mathbb{R})$ since, by (5),

$$\frac{|a - b|}{|\Lambda(\mu, x) - \Lambda(\mu', x')|} \leq \frac{|a - b|}{\|x - x'\|_2 + C\big(W_1(\mu, \mu') - \varepsilon\big)} \leq 1.$$

This completes the proof. $\qquad\square$

## F. Proof of Lemma 13

Fix $(\mu, x), (\mu', x') \in \mathcal{X}$ and $\varepsilon \in (0, 1)$. We recall the Kantorovich–Rubinstein duality:

$$W_1(\mu, \mu') = \sup\Big\{ \int_\Omega \varphi \, d(\mu - \mu') : \ \mathrm{Lip}(\varphi) \leq 1 \Big\}.$$

Fix a point $x_0 \in \Omega$ and consider the normalized class

$$\mathcal{F} := \Big\{ \varphi : \Omega \to \mathbb{R} \ \Big| \ \mathrm{Lip}(\varphi) \leq 1, \ \varphi(x_0) = 0 \Big\}.$$

For any 1-Lipschitz $\varphi$ we may replace it by $\tilde{\varphi} := \varphi - \varphi(x_0)$, which satisfies $\mathrm{Lip}(\tilde{\varphi}) \leq 1$ and $\tilde{\varphi}(x_0) = 0$, and

$$\int \tilde{\varphi} \, d(\mu - \mu') = \int \varphi \, d(\mu - \mu') \quad \text{since } (\mu - \mu')(\Omega) = 0.$$

Hence

$$W_1(\mu, \mu') = \sup_{\varphi \in \mathcal{F}} \int_\Omega \varphi \, d(\mu - \mu').$$

**Step 1: Existence of an optimizer.** Since $\Omega$ is compact, every $\varphi \in \mathcal{F}$ is uniformly bounded: $|\varphi(z)| \leq \mathrm{diam}(\Omega)$ for all $z \in \Omega$. Moreover, $\mathcal{F}$ is equicontinuous and uniformly bounded; thus, by the Arzelà–Ascoli theorem, $\mathcal{F}$ is compact in $(C(\Omega), \|\cdot\|_\infty)$. The map

$$\mathcal{F} \ni \varphi \longmapsto \int_\Omega \varphi \, d(\mu - \mu')$$

is continuous with respect to $\|\cdot\|_\infty$. Therefore there exists $\varphi_0 \in \mathcal{F}$ such that

$$W_1(\mu, \mu') = \int_\Omega \varphi_0 \, d(\mu - \mu').$$

**Step 2: Lipschitz approximation of $\varphi_0$.** By (Murari et al., 2025, Theorem 3.1), for the above $\varphi_0$ there exist a compact set $\Omega_1 \subset \mathbb{R}^h$, a map $Q \in \mathcal{Q}_{d,h,\Omega,\Omega_1}$, a vector $v \in \mathbb{R}^h$, and a 1-Lipschitz map $F : \mathbb{R}^h \to \mathbb{R}^h$ (given as a composition of gradient-descent-type MLPs) such that $v^\top \circ F \circ Q : \mathbb{R}^d \to \mathbb{R}$ is 1-Lipschitz and

$$\sup_{z \in \Omega} \big|\varphi_0(z) - v^\top \circ F \circ Q(z)\big| \leq \varepsilon/2.$$

Consequently,

$$\int_\Omega \varphi_0 \, d(\mu - \mu') \le \int_\Omega v^\top \circ F \circ Q \, d(\mu - \mu') + \varepsilon.$$

Thus

$$W_1(\mu, \mu') \le \int_\Omega v^\top \circ F \circ Q \, d(\mu - \mu') + \varepsilon. \tag{6}$$

**Step 3: Realizing** $\nu \mapsto \int v^\top \circ F \circ Q \, d\nu$ **by one attention layer.** Assume that $F \circ Q \not\equiv 0$ on $\Omega$. Embed $v^\top \circ F \circ Q$ into $\mathbb{R}^{h+1}$ by defining

$$Q_*(z) := \begin{pmatrix} Q(z) \\ 0 \end{pmatrix} \in \mathbb{R}^{h+1}, \qquad F_*(u) := \begin{pmatrix} F(u_{1:h}) \\ 0 \end{pmatrix} \in \mathbb{R}^{h+1}, \qquad \tilde{v} := \begin{pmatrix} v \\ 0 \end{pmatrix} \in \mathbb{R}^{h+1},$$

so that $\tilde{v}^\top \circ F_* \circ Q_*(z) = v^\top \circ F \circ Q(z)$ and $\langle F_* \circ Q_*(z), e_{h+1} \rangle = 0$ for all $z \in \Omega$.

Let

$$A := e_{h+1} \tilde{v}^\top \in \mathbb{R}^{(h+1) \times (h+1)}.$$

Choose

$$\eta := \frac{2}{\sup_{z \in F_* \circ Q_*(\Omega)} \|Az\|_2^2},$$

(then $\eta < \infty$ due to $F \circ Q \not\equiv 0$) so that the attention map below is 1-Lipschitz in the query variable (on the relevant compact set). Define the (gradient-descent-type) attention

$$\Gamma(\nu, z) := z - \eta \int \frac{\exp(\langle z, Aw \rangle)}{\int \exp(\langle z, Aw \rangle) \, d\nu(w)} Aw \, d\nu(w),$$

and let $\bar{\Gamma}$ be the induced in-context map acting on $(\nu, z) \in \mathcal{X}$. Since $Aw = \langle \tilde{v}, w \rangle \, e_{h+1}$, we have

$$\langle F_* \circ Q_*(z), AF_* \circ Q_*(w) \rangle = \langle F_* \circ Q_*(z), e_{h+1} \rangle \langle \tilde{v}, F_* \circ Q_*(w) \rangle = 0,$$

hence the softmax weights become uniform and we obtain

$$\begin{aligned}
&\pi_2 \circ \bar{\Gamma} \circ \bar{F}_* \circ \bar{Q}_*(\nu, z) \\
&= \begin{pmatrix} F \circ Q(z) \\ 0 \end{pmatrix} - \eta \int \frac{e^{\langle F_* \circ Q_*(z), AF_* \circ Q_*(w) \rangle}}{\int e^{\langle F_* \circ Q_*(z), AF_* \circ Q_*(w) \rangle} d\nu(w)} AF_* \circ Q_*(w) d\nu(w) \\
&= \begin{pmatrix} F \circ Q(z) \\ 0 \end{pmatrix} - \eta \int v^\top \circ F \circ Q d\nu \begin{pmatrix} 0 \\ 1 \end{pmatrix} \\
&= \begin{pmatrix} F \circ Q(z) \\ -\eta \int v^\top \circ F \circ Q d\nu \end{pmatrix}.
\end{aligned}$$

Now set $v_* := -\frac{C}{\eta} e_{h+1} \in \mathbb{R}^{h+1}$ and define

$$\Lambda(\nu, z) := v_*^\top \circ \pi_2 \circ \bar{\Gamma} \circ \bar{F}_* \circ \bar{Q}_*(\nu, z) = C \int_\Omega v^\top \circ F \circ Q \, d\nu. \tag{7}$$

By construction, $z \mapsto \Lambda(\nu, z)$ is constant, and $\nu \mapsto \Lambda(\nu, z)$ is $C$-Lipschitz w.r.t. $W_1$. Therefore $\Lambda \in \mathcal{C}_{1,C}(\mathcal{X}, \mathbb{R})$. Moreover, $\Lambda \in \mathcal{K}(\mathcal{X}, \mathbb{R})$, hence $\Lambda \in \mathcal{G}_C(\mathcal{X})$.

So far, Step 3 assumes that $F \circ Q \not\equiv 0$, but the case $F \circ Q \equiv 0$ can be handled by the same argument.

**Step 4: Lower bound.** Using (6) and (7),

$$\Lambda(\mu, x) - \Lambda(\mu', x') = C \int v^\top \circ F \circ Q \, d(\mu - \mu') \ge C\big(W_1(\mu, \mu') - \varepsilon\big).$$

This proves Lemma 13.

$\square$

