# OpenReview forum: "Approximation Theory for Lipschitz Continuous Transformers"
_ICML.cc/2026/Conference — ICML 2026 regular_

### Official Review · Reviewer_urSn · 2026-03-08

**Soundness:** 3
**Presentation:** 2
**Significance:** 3
**Originality:** 2
**Overall Recommendation:** 4
**Confidence:** 3

**Summary:**

This paper investigates the theoretical foundations of Lipschitz continuous Transformers by introducing a novel class of architectures where MLP and attention blocks are formulated as explicit Euler steps of negative gradient flows. The authors adopt a measure-theoretic approach, treating Transformers as operators on probability measures to ensure that approximation guarantees remain independent of the token count. The primary contribution is a universal approximation theorem for 1-Lipschitz in-context maps, supported by a specialized variant of the Restricted Stone-Weierstrass Theorem.

**Compliance With Llm Reviewing Policy:**

Affirmed.

**Final Justification:**

I reconsidered this paper's novelty and found that

(i) its originality might not be so strong (because many ideas of this paper are already known), but
(ii) their Lipschitz-constrained hypothesis class seems to be useful (for future statistical analysis). I think that their integration of existing ideas (negative GF, measure-theoretic settings) is non-trivial.

So I will raise my score, assuming that the authors will revise their draft and clarify their contribution as explained in their rebuttal.

**Key Questions For Authors:**

* Impact of Step Size: Are there inherent trade-offs if the step size $\eta$ or $\tau$ is chosen to be very small? Specifically, does a reduction in step size necessitate a proportional increase in the number of layers $L$ to maintain the same degree of approximation accuracy?
* Potential for Statistical Learning Analysis: Since the current proof is primarily an existence result based on lattice structures, can the authors suggest a path toward a statistical learning framework? Specifically, are there alternative methodologies or related works that could bridge this Lipschitz approximation theory with formal generalization error bounds, which are difficult to derive using the current Stone-Weierstrass-type argument?
* Novelty vs. Furuya et al. (2025): Could the authors elaborate on the specific novelty of their "measure-theoretic in-context formulation" compared to Furuya et al. (2025)? Since the latter already introduced a measure-extension of in-context attention to achieve token-independent guarantees, is the contribution here primarily the integration of this formalism with Lipschitz-constrained architectures rather than the formulation itself?
* Novelty of the GD-type Formulation: Considering that interpreting attention as a gradient-descent-type update is an established concept (e.g., von Oswald et al., 2023 (you cited) and Sander et al. 2022 (you did not cited)), how do the authors characterize the specific novelty of their GD-type attention formulation beyond its role in ensuring Lipschitz continuity?

**Limitations:**

yes

**Strengths And Weaknesses:**

Strengths:
*  Theoretical Rigor and Expressivity: The authors provide a mathematically sound proof demonstrating that each layer of their Euler-step-inspired architecture maintains a 1-Lipschitz constant with respect to the query. Crucially, they show that this structural constraint does not compromise the model's ability to serve as a universal approximator within the Lipschitz-constrained function space.
*  Clarity and Generality: The manuscript is exceptionally well-written and provides a lucid discussion on the Lipschitz properties of attention mechanisms. By formulating the problem within a general measure-theoretic framework, the analysis successfully transcends the limitations of discrete token-based models.

Weaknesses:
* Clarification of Novelty: Certain aspects of the paper’s novelty require further distinction:
    * The "Contributions" section lists the "measure-theoretic in-context formulation" as a key highlight. However, since this formulation appears to be established in prior work (specifically Furuya et al., 2025 ), the authors should more explicitly delineate how their specific implementation or analysis differs from this baseline.
*  Contextualizing Gradient Flow: In the discussion of the gradient flow equation $\dot{x} = -\nabla \lambda(\mu)(x)$ on page 3, it would be appropriate to cite Sander et al. (2022) (e.g., "$\nabla \log $-sum" formulation in Proposition 2 or 3). Referencing their work would help clarify which elements of the gradient flow interpretation are original to this manuscript versus established in existing literature.

* Limitations of the Stone-Weierstrass Approach: The reliance on a variant of the Restricted Stone-Weierstrass Theorem facilitates an existence proof but does not naturally extend to a statistical learning analysis, such as deriving explicit generalization bounds or convergence rates (see Question for further details).

[Sander et al. 2022] Sander, M. E., Ablin, P., Blondel, M., & Peyré, G. (2022, May). Sinkformers: Transformers with doubly stochastic attention. In International Conference on Artificial Intelligence and Statistics (pp. 3515-3530). PMLR.

---

> ### Author Rebuttal · Authors · 2026-03-30
>
> We thank the reviewer for the careful reading and for the constructive and insightful comments. We are particularly grateful for the positive assessment of the theoretical rigor and clarity of the manuscript. We address the main points below and will revise the paper accordingly.
>
> ___
>
> __Impact of Step Size:__
>
>
> We agree that there is a trade-off between step size and depth when interpreting the architecture as a discretization of a gradient flow.
>
> However, we would like to emphasize that, in the approximation setting considered in this paper, the step size does not need to be taken arbitrarily small. In contrast to optimization settings, where stability considerations may force vanishing step sizes, here the depth and step size can be treated as largely independent design parameters, and approximation accuracy does not inherently require the step size to go to zero.
>
> We also note that this perspective is consistent with existing works that adopt a dynamical systems viewpoint for Lipschitz neural networks, where non-vanishing step sizes are used in practice while maintaining stability and strong empirical performance. See, for example:
>
> - Meunier et al.\ (2022), \emph{A dynamical system perspective for Lipschitz neural networks}, ICML,
>
> - Sherry et al.\ (2024), \emph{Designing stable neural networks using convex analysis and ODEs}, Physica D,
>
> - Prach et al.\ (2024), \emph{1-Lipschitz layers compared: Memory speed and certifiable robustness}, CVPR.
>
> We will clarify this point and include these references in the revised manuscript.
>
>
> ___
>
> __Potential for Statistical Learning Analysis:__
>
> We agree that extending the present result toward a statistical learning framework is an important direction.
>
> The current proof is based on a Restricted Stone–Weierstrass-type argument, which yields an existence result but does not provide quantitative approximation rates. Deriving such quantitative estimates in Lipschitz-preserving architectures appears challenging, and we will explicitly acknowledge this limitation in the revised manuscript.
>
> On the other hand, our result is directly relevant to generalization analysis. Since we establish approximation within a class of Lipschitz functions with a controlled constant, the associated hypothesis class admits covering number bounds that have been studied in prior work. This suggests that standard tools from statistical learning theory, such as Rademacher complexity or covering number estimates for Lipschitz function classes, can be applied to obtain generalization guarantees.
>
> We will expand the discussion to clarify this connection.
>
> ___
>
> __Novelty vs. Furuya et al. (2025):__
>
>
> We thank the reviewer for raising this important point.
>
> We would like to clarify that the measure-theoretic in-context formulation itself is not claimed as a novel contribution of this work. Rather, our contribution is __to integrate this formulation with architectural Lipschitz constraints, and to establish a universal approximation theorem within this constrained class.__
>
> More specifically, the universality result in Furuya et al. (2025) is established for unconstrained Transformer architectures, whereas our result shows that universality can be retained even under explicit Lipschitz-preserving architectural constraints.
>
> We will revise the manuscript to make this distinction more explicit.
>
> ___
>
> __Novelty of the GD-type Formulation:__
>
>
> We agree that interpreting attention mechanisms through gradient-flow or optimization perspectives has been explored in prior work (e.g., von Oswald et al., Sander et al.).
>
> The novelty of our approach lies not in introducing this perspective itself, but in using it as a systematic architectural principle to enforce Lipschitz continuity, while preserving universal approximation in the in-context setting. More explicitly, our main contribution is the approximation theory for these gradient-based architectures, rather than the architecture itself.
>
> We will include the suggested reference (Sander et al., 2022) and clarify this positioning in the revision.

---

> > ### Author Rebuttal · Reviewer_urSn · 2026-04-02
> >
> > Thank you for your explanation.
> > I reconsidered this paper's novelty and found that
> >
> > - (i) its originality might not be so strong, but
> > - (ii) their Lipschitz-constrained hypothesis class seems to be useful (for future statistical analysis) and non-trivial.
> >
> > I will raise my score, assuming that the authors will revise their draft and clarify their contribution as explained in their rebuttal.

---

### Official Review · Reviewer_4Ndq · 2026-03-13

**Soundness:** 3
**Presentation:** 3
**Significance:** 3
**Originality:** 3
**Overall Recommendation:** 5
**Confidence:** 1

**Summary:**

The paper addresses a existing gap in transformer theory by proposing a Lipschitz-continuous transformer architecture and establishing universal approximation guarantees, by treating inputs as general probability measures. The authors propose a gradient-descent-like transformer where both MLP and attention blocks are formulated as explicit Euler steps of negative gradient flows. The design ensures that the resulting map is 1-Lipschitz with respect to the query variable and context variables. The paper establishes approximation guarantees independent of token count. The paper is purely theoretical and omits any experimental evaluations due to the difficulties, the authors discuss in the conclusion/discussion section.

**Compliance With Llm Reviewing Policy:**

Affirmed.

**Final Justification:**

The authors have addressed most of my concerns -- their results seem solid and pave the way for the development of more advanced transformer architectures.

**Key Questions For Authors:**

See questions in the weakness sections above.

**Limitations:**

yes

**Strengths And Weaknesses:**

## Strengths

The paper makes several notable contributions:

- it introduces a novel architectural paradigm by casting transformers as gradient flow iterations, which allows for Lipschitz control.
- the theoretical analysis seems rigorous, providing the first universal approximation theorem for Lipschitz-constrained transformers. The measure-theoretic formulation allows to handle variable-length contexts unlike the general discrete empirical measure case.

## Weaknesses

- the authors explicitely state that the step size $\eta_\ell$ in attention layers depends on $\sup_{y \in \Omega_\ell} \|A y\|_2^2$, which as far as I understand, can be estimated via power method or Gelfand's formula, however it obviously increases computational burden.
- while the theoretical part seems interesting on its own, a more elaborate discussion on its implications for transformer design and applicability to practical settings could benefit the paper.


Not being a specialist in the field, I can't fully understand the novelty and impact of the papers's contributions.

---

> ### Author Rebuttal · Authors · 2026-03-30
>
> We thank the reviewer for the positive assessment and for highlighting the main contributions of the paper. We are particularly encouraged that the theoretical results and the measure-theoretic formulation are appreciated. We address the points below and will revise the manuscript to improve clarity and positioning.
>
> ___
>
> >the authors explicitely state that the step size ...
>
> We agree that the admissible step sizes in attention layers depend on layerwise quantities of the form $\sup_{y\in\Omega_\ell}\|Ay\|_2^2$, which in practice may require estimating both operator norms and the relevant input domains, potentially increasing computational cost.
>
> We would like to emphasize that the goal of this work is to establish a theoretical existence result under explicit architectural Lipschitz constraints, rather than to provide an immediately optimized implementation.
>
> We will expand the discussion to clarify:
>
> - how such quantities may be estimated in practice,
>
> - and that this introduces additional computational considerations, which we acknowledge as a limitation.
>
> ___
>
> >while the theoretical part seems interesting on its own ...
>
> We agree that a more detailed discussion of practical implications would improve the paper.
>
> The main message of this work is that:
>
> __it is possible to enforce Lipschitz constraints at the architectural level without sacrificing universal approximation power,
> in a setting that is independent of the number of tokens.__
>
> This provides a conceptual foundation for designing stable and robust Transformer variants.
>
> The present work is theoretical, and it sets the foundations for more practical extensions focusing on aspects such as:
>
> - designing efficiently implementable Lipschitz-controlled attention mechanisms,
>
> - avoiding explicit norm estimation via alternative parameterizations,
>
> - and connecting with existing normalization or stable Transformer variants.
>
>
> We will expand the discussion section to better articulate these implications.
>
> ___
>
> >Not being a specialist in the field, I can't fully understand the novelty and impact ....
>
>
> We appreciate the reviewer’s comment and will clarify the novelty more explicitly.
>
> Our contribution is not to study approximation in unconstrained Transformer classes. Rather, we show that:
>
>
> __universal approximation holds within a class of architectures that enforce Lipschitz continuity by design,
> in a measure-theoretic in-context setting that is independent of token count.__
>
> This differs from:
>
> - standard Transformer approximation results, where Lipschitz constants are not controlled and typically grow with model complexity,
>
> - and prior measure-theoretic formulations, which do not incorporate architectural Lipschitz constraints.
>
> We will revise the manuscript to make this distinction clearer.

---

> > ### Author Rebuttal · Reviewer_4Ndq · 2026-04-02
> >
> > I thank the authors for clarifications, and have no further questions regarding the paper.

---

### Official Review · Reviewer_jFYm · 2026-03-13

**Soundness:** 2
**Presentation:** 3
**Significance:** 1
**Originality:** 2
**Overall Recommendation:** 2
**Confidence:** 5

**Summary:**

The paper studies Transformer architectures and looks at potential formulations that are Lipschitz continuous. The authors claim to design continuous Transformer layers where both MLP and attention blocks are interpreted as forward Euler steps of gradient flows. This design allows control of the Lipschitz constant of the model by construction. The contribution is a universal approximation theorem for this class of Lipschitz Transformers, where the context tokens are represented as a probability measure rather than a finite sequence. The authors prove that this architecture can approximate any Lipschitz in-context mapping under some constraints.

**Compliance With Llm Reviewing Policy:**

Affirmed.

**Key Questions For Authors:**

1) Since negative gradient flows are already known to produce 1-Lipschitz mappings, what is really new compared to previous work on gradient-flow and Lipschitz neural networks?

2) The stability guarantees are only established with respect to the query token while the context is treated as fixed. Transformers strongly depend on the context. Why is the Lipschitz analysis limited to the query and not the context?

3) The theorem does not show how large or deep a network must be to reach good accuracy or whether enforcing Lipschitz constraints limits modeling power. How efficient is the approximation in practice?

**Limitations:**

Yes !

**Strengths And Weaknesses:**

## Strengths
Designing Lipschitz continuous Transformer blocks is an important missing component relevant for robustness, stability, and adversarial guarantees.

The paper is rigorous and detailed.

The paper introduces a clean formulation of Transformers as Lipschitz operators acting on a context–query pair, where the context is represented as a probability measure.

## Weakness
W1) “We realize both MLP and attention blocks as explicit Euler steps of negative gradient flows, ensuring inherent stability without sacrificing expressivity.”

The abstract claim is relatively strong, as similar ideas have appeared in prior works cited by the authors.
Negative gradient flows are known to yield 1-Lipschitz mappings.

(Meunier et al., 2022) Dynamical System Perspective for Lipschitz Neural Networks (the correct date is 2022).

Related ideas also appear in more recent work closely related to Lipschitz attention variants, such as

Globally 1-Lipschitz Attention Without Sequence-Length Dependence, 2026. URL https://openreview.net/forum?id=9Y7L5VeV4Z. Under review at the International Conference on Learning Representations (ICLR).

It may therefore be helpful to adjust the claim to position the contribution more clearly with respect to these past works. However, the current work appears complementary, as it focuses on the approximation theory of Lipschitz Transformers rather than on architecture construction.

W2) The Lipschitz analysis is established only with respect to the query variable, while the context is treated as fixed. This somewhat limits the scope of the stability analysis, since the sensitivity of the model with respect to changes in the context is not addressed.

W3) Modeling the context as a probability measure is a convenient abstraction, but it removes dependence on the number of tokens and the precise ordering of tokens, which may overlook important sequential dependencies that real Transformer models rely on. In practice, the worst-case Lipschitz bound of Transformers can scale with sequence length due to attention interactions. The present analysis removes this dependence and therefore ignores this important aspect of Transformer behavior.

W4) The theorem establishes that the Lipschitz architecture remains universal, but it does not provide quantitative bounds on the network size required to achieve a given accuracy, so the efficiency of the representation remains unclear.

W5) The fact that no empirical setup is proposed (even a toy one) makes me wonder whether the proposed method is actually usable. Furthermore, the lack of experimental illustration makes the goal/motivation of the authors a bit unclear.

---

> ### Author Rebuttal · Authors · 2026-03-30
>
> We thank the reviewer for the detailed and careful assessment, as well as for recognizing the importance of Lipschitz-continuous Transformer architectures. We address the main concerns below and will revise the manuscript to clarify the novelty and scope of our contribution.
>
> ___
>
> __W1 and Q1: On novelty relative to gradient-flow and Lipschitz neural networks__
>
> We agree that the use of negative gradient flows to construct 1-Lipschitz mappings is well established in prior work. Our contribution is not to introduce this principle itself.
>
> Rather, the key novelty of this paper is the following:
>
> __we show that universal approximation is achievable within a class of Transformer architectures that enforce Lipschitz continuity by design, in an in-context setting where the context is modeled as a probability measure and the guarantees are independent of the number of tokens.__
>
> In contrast:
>
> - prior work on Lipschitz neural networks focuses on architecture design and stability
>
> - prior Transformer approximation results do not enforce Lipschitz constraints at the architectural level and typically do not preserve them during approximation
>
> We will revise the manuscript to clarify this positioning and adjust the claims in the abstract accordingly.
>
> ___
>
> __W2 and Q2: On Lipschitz continuity with respect to the context__
>
> We would like to clarify that the context is not treated as fixed in our analysis.
>
> In our setting, the target class consists of functions that are 1-Lipschitz with respect to the query variable and $C$-Lipschitz with respect to the context variable, for an arbitrary fixed constant $C>0$.
>
> Accordingly, our approximation result establishes that such joint Lipschitz mappings can be approximated within the proposed architecture, preserving the Lipschitz structure in both variables.
>
> We will revise the manuscript to make this point more explicit and avoid possible confusion.
>
> ___
>
> __W3: On modeling context as a probability measure__
>
> We agree that sequence-based formulations that explicitly account for token order and sequence length are important, and provide a complementary perspective.
>
> However, our goal is different: we study approximation in a regime where the number of tokens is not fixed a priori.
>
> In standard sequence-based analyses developed in prior works, approximation typically involves model complexity that depends on the number of tokens, and empirical observations suggest that increasing token count may lead to increased parameter size or Lipschitz constants.
>
> In contrast, by formulating the problem on the space of probability measures and leveraging a Stone–Weierstrass-type argument, we show that approximation can be achieved with a finite number of parameters even in a setting that effectively corresponds to arbitrarily many tokens.
>
> Moreover, our result establishes that this can be done while maintaining explicit Lipschitz control that does not depend on the number of tokens or the target accuracy.
>
> We will clarify this modeling perspective and its implications in the revised manuscript.
>
> ___
>
> __W4 + Q3: On quantitative efficiency and approximation rates__
>
> We agree that the current result does not provide explicit bounds on network size or depth required to achieve a given accuracy.
>
> This is due to the use of a Restricted Stone–Weierstrass-type argument, which yields an existence result but is inherently non-quantitative.
>
> Deriving approximation rates under architectural Lipschitz constraints remains an important open problem. We will explicitly acknowledge this limitation and discuss possible directions for obtaining quantitative bounds.
>
> ___
>
> __W5: On the lack of empirical evaluation__
>
> The purpose of this work is to __establish a structural result: that architectural Lipschitz constraints can be enforced without sacrificing universal approximation, and we provide such an analysis in a general token-count-independent, measure-theoretic setting.__
> We view the development of practically implementable and certifiable architectures satisfying these conditions as an important direction for future work, and we will clarify this perspective in the discussion section.

---

> > ### Author Rebuttal · Reviewer_jFYm · 2026-04-03
> >
> > I thank the authors for the answers, however many of them do not really address the weaknesses I raised. The paper contains only 7 pages and the page left could bring the opportunity to add some experimental results to illustrate whether the findings do apply (even on toy or synthetic data).

---

> > > ### Author Response · Authors · 2026-04-05
> > >
> > > >The paper contains only 7 pages and the page left could bring the opportunity to add some experimental results to illustrate whether the findings do apply (even on toy or synthetic data).
> > >
> > > We thank the reviewer for this important comment. We conducted additional experiments on a simple synthetic 1D in-context regression task.
> > >
> > > We consider the following setup. Let
> > > $$
> > > f_a(x) = a \tanh x,
> > > \qquad |a| \le 1,
> > > $$
> > > $$
> > > g_a(x) = a \sin x,
> > > \qquad |a| \le 1,
> > > $$
> > > $$
> > > h_{a,b}(x) = a \sin x + \frac{b}{2} \sin(2x),
> > > \qquad |a| + |b| \le 1,
> > > $$
> > > which define families of $1$-Lipschitz functions parameterized by $a$ or $(a,b)$.
> > >
> > > In each episode, we sample a small set of context points $(x_i, y_i)$ given by
> > > $$
> > > y_i = f_a(x_i) + \xi_i,
> > > \qquad
> > > y_i = g_{a}(x_i) + \xi_i,
> > > \qquad
> > > y_i = h_{a,b}(x_i) + \xi_i,
> > > $$
> > > where $\xi_i$ denotes additive noise. The goal is to predict $f_a(x_q)$ or $g_{a}(x_q)$ or $h_{a,b}(x_q)$ at a query point $x_q$ from the noisy observations $(x_i, y_i)_{i=1}^n$.
> > >
> > > We compare the following two models:
> > >
> > > - __Proposed method:__ a transformer based on the gradient-flow formulation in Section~3.
> > > - __Baseline:__ a standard transformer with the same total number of trainable parameters.
> > >
> > > For the tasks involving $f_a$ and $g_a$, we use a one-layer transformer, while for $h_{a,b}$ we use a two-layer transformer. For fairness, we match the total number of trainable parameters between the models.
> > > For fairness, we match the total number of trainable parameters between the models.
> > >
> > > In the proposed method, each block is implemented as a single explicit Euler step of a gradient flow. For the attention block, the update takes the form
> > > $$
> > > x \mapsto x - \eta \, A^\top \mathrm{softmax}(A\,\cdot),
> > > $$
> > > with step size
> > > $$
> > > \eta = \frac{2}{\max_j \|A c_j\|^2 + \varepsilon},
> > > $$
> > > where $c_j$ denotes context tokens. Similarly, for the MLP block, we use
> > > $$
> > > x \mapsto x - \tau \, W^\top \mathrm{ReLU}(Wx+b),
> > > \qquad
> > > \tau = \frac{2}{\|W\|_{\mathrm{op}}^2 + \varepsilon}.
> > > $$
> > > Thus, the step sizes are normalized by the corresponding operator magnitudes, consistent with the non-expansive / Lipschitz-controlled structure described in Section~3.
> > >
> > > In addition to prediction accuracy (MSE/MAE), we evaluate stability via sensitivity metrics. Denoting the model output by $F(\mathcal C, x_q)$, where $\mathcal C$ is the context set, we define the query sensitivity and context sensitivity as
> > > $$
> > > S_{\mathrm{query}}
> > > :=
> > > \frac{|F(\mathcal C, x_q+\delta x_q) - F(\mathcal C, x_q)|}{|\delta x_q|}
> > > , \quad
> > > S_{\mathrm{context}}
> > > :=
> > > \frac{|F(\mathcal C + \delta \mathcal C, x_q) - F(\mathcal C, x_q)|}{\|\delta \mathcal C\|}.
> > > $$
> > >
> > > The results are summarized as follows :
> > >
> > > ___
> > >
> > > __True target $f_a(x)=a \tanh(x)$__
> > > | Model | #Params | Test MSE | Test MAE  | Mean joint sensitivity  | Mean context-only sensitivity  | Mean query-only sensitivity  |
> > > |---|---:|---:|---:|---:|---:|---:|
> > > | Proposed method | 201 | 0.032056 | 0.137474 | 0.106427 | 0.156752 | 0.094183 |
> > > | Baseline | 201 | 0.020317 | 0.105542 | 0.111785 | 0.171054 | 0.110465 |
> > >
> > > ___
> > >
> > > __True target $g_a(x)=a \sin(x)$__
> > > | Model | #Params | Test MSE | Test MAE  | Mean joint sensitivity  | Mean context-only sensitivity  | Mean query-only sensitivity  |
> > > |---|---:|---:|---:|---:|---:|---:|
> > > | Proposed method | 201 | 0.050410 | 0.167053 | 0.117016 | 0.147993 | 0.228674 |
> > > | Baseline | 201 | 0.039302 | 0.138268 | 0.117983 | 0.147020 | 0.295830 |
> > >
> > > ___
> > >
> > > __True target $h(x) = a \sin x + \frac{b}{2} \sin(2x)$__
> > > | Model | #Params | Test MSE | Test MAE  | Mean joint sensitivity  | Mean context-only sensitivity  | Mean query-only sensitivity  |
> > > |---|---:|---:|---:|---:|---:|---:|
> > > | Proposed method | 399 | 0.040690 | 0.154963 | 0.087939 | 0.698648 | 0.248997 |
> > > | Baseline | 399 | 0.029278 | 0.130067 | 0.098450 | 0.893820 | 0.287273 |
> > >
> > >
> > > ___
> > >
> > > The results show that:
> > >
> > > - The baseline achieves lower MSE and MAE than the proposed method.
> > >
> > > - The proposed method consistently exhibits lower sensitivity, particularly in the query direction.
> > >
> > > In particular, __the reduction in query sensitivity is consistently significant across all three tasks__. Although visualizations cannot be included in the rebuttal, we observe that the proposed method produces smoother approximations of $f_a$, $g_a$, and $h_{a,b}$, whereas the baseline sometimes yields oscillatory or irregular predictions.
> > >
> > > This behavior aligns with our theoretical findings. The proposed architecture explicitly controls the Lipschitz constant, especially in the query direction, which is crucial in in-context regression tasks where stable approximation of families of $1$-Lipschitz functions is required.
> > >
> > > Overall, these results suggest that the Lipschitz control and non-expansive structure induced by the gradient-flow formulation translate into improved stability in practice. While the current experiment is conducted in a simplified setting, it provides empirical support for the theoretical properties established in the paper.
> > >
> > > Further experimental details will be included in the revised version.

---

### Official Review · Reviewer_4hVn · 2026-03-16

**Soundness:** 2
**Presentation:** 2
**Significance:** 2
**Originality:** 2
**Overall Recommendation:** 4
**Confidence:** 4

**Summary:**

This paper proposes a class of Lipschitz-preserving in-context Transformers in which both the MLP block and the attention block are interpreted as explicit Euler steps of negative gradient flows.
The model is formulated in a measure-theoretic in-context setting: the context is treated as a probability measure, the query is treated as a point, and the goal is to obtain maps that are 1-Lipschitz in the query and C-Lipschitz in the context.
The main theorem states that for any epsilon in (0,1) and any scalar-valued target Lambda in C_{1,C}(X, R), there exists Lambda in G_C(X, R) that uniformly approximates Lambda.
The proof is based on a variant of the Restricted Stone-Weierstrass theorem, with lattice and separation properties established through Lemmas 10-13.
The paper positions this as the first approximation-theoretic guarantee for Transformer architectures that explicitly preserve Lipschitz continuity.

**Compliance With Llm Reviewing Policy:**

Affirmed.

**Final Justification:**

My concerns have been resolved, so I raise my score.

**Key Questions For Authors:**

1. In Lemma 12, should the affine-rescaling coefficient be stated as |alpha| <= 1 rather than alpha in [0,1]? If signed affine maps are intended to be representable in K(X, R), please make this explicit and explain how the construction works in the case alpha < 0.

2. Which parts of the analysis fundamentally require the constraint V = A? More specifically, what breaks if one allows independently parameterized Q, K, V, or standard multi-head attention?

3. Is it possible to provide any approximation-complexity bound, even a very coarse one, in terms of epsilon, lifting dimension, or depth? Without such a bound, it is hard to judge whether the universality theorem has any implication for realistic model sizes.

4. Could the authors say more about the path toward vector-valued outputs? The current discussion states that this extension is nontrivial; it would be helpful to know whether the obstacle is mainly technical or whether the current proof strategy is fundamentally scalar-specific.

5. Empirical analysis would strengthen the paper considerably. Even a synthetic experiment showing that the proposed architecture does not collapse expressivity in practice would make the theoretical message much more convincing.

6. Could the authors  state explicitly that the contribution is not approximation of Lipschitz functions, but approximation under explicit architectural Lipschitz control? This would better separate the paper from prior standard-Transformer approximation results and from earlier measure-theoretic in-context universality results.

7. Do any of the existing standard-Transformer approximation constructions already satisfy a fixed Lipschitz budget that is independent of epsilon? If not, could the authors explain precisely why those results do not imply the present theorem? This seems central to the paper’s novelty.

8. Could the authors compare their model class more directly to the standard-Transformer results that already derive approximation rates and Lipschitz upper bounds? A short discussion clarifying standard Transformer with epsilon-dependent complexity versus architecturally Lipschitz-preserving Transformer clas” would make the contribution much easier to evaluate

**Limitations:**

See Weaknesses and Questions

**Strengths And Weaknesses:**

Strengths

1. The architecture and the theory are internally aligned.
The attention layer is redesigned as a gradient-descent-type update, and the measure-theoretic formulation gives token-count-independent guarantees.
This makes the paper mathematically coherent rather than a collection of disconnected technical tricks.

2. The paper uses a Restricted Stone-Weierstrass style argument and explicitly constructs the approximation through lattice operations.
In particular, the appendix shows the finite min/max construction underlying the density argument, which gives useful structural insight into why universality holds in this model class.

Weaknesses

1. The theorem is proved for scalar-valued targets in C_{1,C}(X, R), not for vector-valued outputs.
The discussion explicitly acknowledges that extending the result to vector-valued outputs is nontrivial and likely requires new approximation mechanisms beyond the scalar proof strategy.
This substantially limits how directly one can interpret the result for standard Transformer representation learning settings.

2.The discussion explicitly notes that there remains a gap between the abstract universality theorem and practical, certifiable architectures.
In particular, the admissible step size eta_l depends on the layerwise input domain Omega_l, which is difficult to estimate in practice, and the Lipschitz bound for in-context attention is expressed through a supremum quantity that may be hard to estimate or certify.
This makes the result more of a foundational feasibility theorem than a practically actionable recipe.

3.The architecture is more specialized than a standard Transformer.
The proposed attention layer fixes V = A, and the paper explicitly says this constraint is introduced to enable the gradient-flow interpretation and plays an essential role in the proof of Lemma 2(1).
As a result, it is not clear how much of the theory transfers to independently parameterized Q, K, V or to standard multi-head attention.

4. The claimed “explicit Lipschitz control” is not fully parameter-level.
The hypothesis class is defined as G_C(X, R) = C_{1,C}(X, R) intersect K(X, R), and the paper explicitly explains that this intersection is needed because the authors do not impose norm constraints on v or Q.
Mathematically this is legitimate, but it weakens the implementation-level interpretation: the theorem is not stated for a cleanly parameter-constrained model class alone, but for the subset of representable functions that also satisfy the desired Lipschitz constraints.

5. There are no quantitative complexity bounds. The theorem is existential and does not provide bounds on the required lifting dimension, width, depth, or parameter count as a function of approximation error. Therefore, the result does not show that realistic-size Lipschitz-constrained Transformers retain high expressivity; it only shows that some model in the class can approximate the target function.

6. The paper emphasizes that the proof is constructive, but this should be interpreted carefully.
The Stone-Weierstrass part is constructive in the sense of explicit min/max compositions, but key parts of the argument still rely on existence results such as Arzela-Ascoli for an optimizer in the Kantorovich-Rubinstein dual formulation and a prior universal approximation theorem for 1-Lipschitz MLPs.
Thus, the proof is not a practical synthesis algorithm that turns an arbitrary target directly into concrete weights with efficiency guarantees.

7. I also see one technical point that should be clarified carefully. In the proof of Lemma 12, the final affine rescaling writes Lambdahat = alpha * Lambda + beta and states alpha in [0,1].
As written, the construction seems to justify only |alpha| <= 1, since alpha can be negative when a < b. This may be repairable if signed affine maps are indeed representable within the model class, but in its current form it reads like a proof gap or at least a notation/argument mismatch that should be fixed before publication.

8. The paper should make the distinction between target-side smoothness and model-side architectural Lipschitz control much more explicit. The earlier rate papers prove approximation for standard Transformers on Hölder classes, while the current paper claims approximation-theoretic guarantees for architectures that explicitly preserve Lipschitz continuity. Those are different objects, but the current framing risks blurring them.

Transformers are Universal In-context Learners
Takashi Furuya, Maarten V. de Hoop, Gabriel Peyré

Approximation Rate of the Transformer Architecture for Sequence Modeling
Haotian Jiang, Qianxiao Li

---

> ### Author Rebuttal · Authors · 2026-03-30
>
> We thank the reviewer for the careful and insightful evaluation.
>
> ### Clarification of contribution.
> Our contribution is not an approximation theory for Lipschitz functions in the unconstrained sense. Rather, **we show that universal approximation remains possible within a class of Transformer architectures that enforce Lipschitz continuity by design.** This is done in a measure-theoretic in-context setting that controls both query and context dependence and is independent of token count. This distinguishes our result from prior approximation results for standard, unconstrained Transformers.
>
> We address below the key questions pointed out in the review.
> ___
>
> ### Key Questions
>
> **Q1.** The reviewer is correct that $\alpha \in [0,1]$ should be replaced by $|\alpha| \leq 1$ and signed affine transformations are representable in our class. We will correct this in the revision.
> ___
>
> **Q2.** The constraint $V = A$ is introduced to enable the gradient-flow interpretation and is used in Lemma 2(1).
>
> While more general parameterizations (e.g., independent $Q,K,V$ or multi-head attention) can represent this case, our focus is not expressivity but enforcing Lipschitz control. The constraint $V = A$ is essential for guaranteeing this stability; without it, such control is unclear.
>
> We will clarify in the revision which parts of the analysis are tied to this constraint and discuss possible extensions to more general $Q,K,V$ parameterizations.
>
> ___
>
> **Q3.** The current result is an existence theorem based on a Restricted Stone–Weierstrass-type argument, which is inherently non-quantitative. Deriving explicit bounds in terms of $\varepsilon$, lifting dimension, or depth would require different techniques, and we will clarify this limitation and discuss it as future work. At the same time, we emphasize that this existence result under architectural Lipschitz constraints is an important first step toward a general approximation theory for Lipschitz Transformers, even without quantitative rates.
>
> ___
>
> **Q4.** We agree that extending the result to vector-valued outputs is an important direction.
>
> For output spaces equipped with the $\ell^\infty$ norm, the extension is straightforward: one can apply the scalar approximation argument coordinate-wise and take the maximum over coordinates. We will clarify this in the revision.
> In contrast, equipping the output space with $\ell^2$ leads to a substantially more delicate setting. The main difficulty is structural: the class of vector-valued $1$-Lipschitz functions (in $\ell^2$) is not a lattice, so the key min/max-based argument does not extend.
>
> Moreover, componentwise reasoning is inherently limited in the $\ell^2$-valued setting. If one controls each coordinate separately, the resulting vector-valued map is at best bounded by a coarse $\sqrt{d}$ Lipschitz estimate in $\ell^2$. This bound is generally not sharp. For example, the identity map on $\mathbb{R}^d$ has coordinates that are each exactly $1$-Lipschitz, while the full map is still $1$-Lipschitz in $\ell^2$, not $\sqrt{d}$-Lipschitz. Thus, coordinatewise arguments do not capture the true geometry of the vector-valued problem and the extension beyond scalar functions requires new ideas even in the context-free case.
>
> ___
>
> **Q5.** See W5 "On the lack of empirical evaluation." in Reviewer jFYm.
>
> ___
>
> **Q6.** Lemma 13 relies on Arzelà–Ascoli to assert the existence of a maximizer $\phi_0$, so the argument is not fully constructive in this sense. We will revise the manuscript to make this explicit.
>
> At the same time, an important constructive aspect remains. The Restricted Stone–Weierstrass step yields explicit approximants via minimax compositions, and the scalar $1$-Lipschitz ResNet universality result provides a concrete realization once $\phi_0$ is given. Thus, the only genuinely nonconstructive ingredient is the use of Kantorovich–Rubinstein duality; conditional on this, the subsequent steps are constructive.
>
> We will clarify that the argument is constructive modulo the choice of the dual optimizer, and that the network realization is explicit.
>
> ___
>
> **Q7.** Existing standard-Transformer approximation results do not provide constructions with Lipschitz constants bounded independently of $\varepsilon$. As $\varepsilon \to 0$, model complexity typically increases, and so does the Lipschitz constant. Moreover, approximation in sup norm alone does not control Lipschitz constants (for instance, $F(x)=T(x)+\varepsilon\sin(nx)$ is uniformly $\varepsilon$-close to $T$, while its Lipschitz constant can be made arbitrarily large by increasing $n$).
>
> ___
>
> **Q8.** Both approaches require $\varepsilon$-dependent complexity, but differ fundamentally in stability:
>
> Standard Transformers: no uniform Lipschitz control
>
> Our model: $\varepsilon$-independent Lipschitz guarantees enforced at the architectural level, and independence from token count
>
> We will revise the manuscript to make this distinction more explicit.

---

> > ### Author Rebuttal · Reviewer_4hVn · 2026-04-03
> >
> > Thank you for your response. My concerns have been resolved, so I will raise my score.

---

### Decision · Program_Chairs · 2026-04-30

**Decision:**

Accept (regular)

**Comment:**

This paper theoretically investigates the approximation ability of multi-layer Transformers to approximate Lipschitz continuous functions where the input consists of a measure and a query vector. The authors employ the measure-theoretic formulation of Transformer and the Lipschitz continuity is defined with respect to 1-Wasserstein distance for the measure input and Euclidean distance for the vector input respectively. The key ingredient of the proof is a modification of the Restricted Stone–Weierstrass Theorem, and the authors successfully showed the universal approximation ability even with a Lipschitz continuity constraint (on both the model and the target function).

The main contribution of this paper is on the establishment of the Restricted Stone–Weierstrass Theorem and its application to the Transformer architecture, which is novel. The writing is overall clear. The authors clearly explain each mathematical concept and the theoretical statement and its explanations are rigorously described. On the other hand, there are few drawbacks:
1. The Lipschitz continuity constraint is given with respect to a function level instead of the parameter level. This is a bit weak from a practical point of view.
2. Some reviewers pointed out its novelty compared to existing work is not adequately described. For example, connection to Furuya et al. (2025) should be described appropriately. Overall, these points were addressed during the rebuttal period, the authors should reflect this correction to the manuscript.
3. The approximation theory is not quantitative as inherited from Stone–Weierstrass type argument.

Although there are such drawbacks as I stated above, this submission provides a clear description of the universal approximation theory. This is valuable to the community. The reviewers appreciate the contribution after the rebuttals. Then, I recommend acceptance.